# The relationship between psychosocial circumstances and injuries in adolescents: An analysis of 87,269 individuals from 26 countries using the Global School-based Student Health Survey

Samiha Ismail[1,2,3☯], Maria Lisa Odland[3☯], Amman Malik[4], Misghina Weldegiorgis[5,6], Karen Newbigging[7], Margaret Peden[5], Mark Woodward[5,6], Justine Davies[3,8,9]*

1 Centre for Medical Education, Institute of Health Sciences Education, Queen Mary University of London, London, United Kingdom, 2 University College London Hospital, London, United Kingdom, 3 Institute for Applied Health Research, University of Birmingham, Birmingham, United Kingdom, 4 Medical School, College of Medical and Dental Sciences, University of Birmingham, United Kingdom, 5 The George Institute for Global Health, School of Public Health, Imperial College London, United Kingdom, 6 The George Institute for Global Health, University of New South Wales Sydney, Sydney, Australia, 7 School of Social Policy and Institute for Mental Health, University of Birmingham, United Kingdom, 8 Medical Research Council/Wits University Rural Public Health and Health Transitions Research Unit, Faculty of Health Sciences, School of Public Health, University of the Witwatersrand, Johannesburg, South Africa, 9 Centre for Global Surgery, Department of Global Health, Stellenbosch University, Stellenbosch, South Africa

☯ These authors contributed equally to this work.
* j.davies.6@bham.ac.uk

**Data Availability Statement:** This study conducted a secondary analysis of data collected from the

## Abstract

### Background

Over a million adolescents die globally each year from preventable or treatable causes, with injuries (intentional and unintentional) being the leading cause of these deaths. To inform strategies to prevent these injuries, we aimed to assess psychosocial factors associated with serious injury occurrence, type, and mechanism in adolescents.

### Methods and findings

We conducted a secondary analysis of cross-sectional survey data collected from the Global School-based Student Health Survey between 2009 and 2015. We used logistic regression to estimate associations between prevalence of serious injuries, injury type (effects of injury), and injury mechanism (cause of injury) and psychosocial factors (factors that relate to individuals socially, or their thoughts or behaviour, or the interrelation between these variables). Psychosocial factors were categorised, based on review of the literature, author knowledge, and discussion amongst authors. The categories were markers of risky behaviour (smoking, alcohol use, drug use, and physical activity), contextual factors (hunger, bullying, and loneliness), protective factors (number of friends and having a supportive family), and markers of poor mental health (planned or attempted suicide and being too worried to sleep). Models were adjusted for country factors (geographical area and income

Global Schools-Based Student Health Survey. Module questions and data from this survey are freely available for download and analysis (www. cdc.gov/GSHS/ and www.who.int/ncds/ surveillance/gshs/).

**Funding:** The authors received no specific funding for this work.

**Competing interests:** I have read the journal's policy and the authors of this manuscript have the following competing interests: M.P. was a WHO staff member when the school behaviour survey tools were developed and had input into the injury module, she is no longer employed by WHO. We declare no other competing interests.

**Abbreviations:** GSHS, Global School-based Student Health Survey; LMICs, low- and middle-income countries; MVA, motor vehicle accident; OR, odds ratio; PSU, primary sampling unit; SDG, Sustainable Development Goal; WHO, World Health Organization.

status, both using World Bank classification), demographic factors (age and sex), and factors to explain the survey design. A total of 87,269 adolescents living in 26 countries were included. The weighted majority were 14–15 years old (45.88%), male (50.70%), from a lower-middle-income country (81.93%), and from East Asia and the Pacific (66.83%). The weighted prevalence of a serious injury in the last 12 months was 36.33%, with the rate being higher in low-income countries compared to other countries (48.74% versus 36.14%) and amongst males compared to females (42.62% versus 29.87%). Psychosocial factors most strongly associated with serious injury were being bullied (odds ratio [OR] 2.45, 95% CI 1.93 to 3.13, $p < 0.001$), drug use (OR 2.08, 95% CI 1.73 to 2.49, $p < 0.001$), attempting suicide (OR 1.78, CI 1.55 to 2.04, $p < 0.001$), being too worried to sleep (OR 1.80, 95% CI 1.54 to 2.10, $p < 0.001$), feeling lonely (OR 1.61, 95% CI 1.37 to 1.89, $p < 0.001$), and going hungry (OR 1.61, 95% CI 1.30 to 2.01, $p < 0.001$). Factors hypothesised to be protective were not associated with reduced odds of serious injury: Number of close friends was associated with an increased odds of injury (OR 1.23, 95% CI 1.06 to 1.43, $p = 0.007$), as was having understanding parents or guardians (OR 1.13, 95% CI 1.01 to 1.26, $p = 0.036$). Being bullied, using drugs, and attempting suicide were associated with most types of injury, and being bullied or too worried to sleep were associated with most mechanisms of injury; other psychosocial factors were variably associated with injury type and mechanism. Limitations include the cross-sectional study design, making it not possible to determine the directionality of the associations found, and the survey not capturing children who did not go to school.

## Conclusions

We observed strong associations between serious injury and psychosocial factors, but we note the relationships are likely to be complex and our findings do not inform causality. Nevertheless, our findings suggest that multifactorial programmes to target psychosocial factors might reduce the number of serious injuries in adolescents, in particular programmes concentrating on reducing bullying and drug use and improving mental health.

## Author summary

### Why was this study done?

- Globally, the leading cause of disability-adjusted life years in adolescents is road traffic injuries (self-harm and interpersonal violence are also in the top 5), and there is an urgent need to reduce this preventable burden of injuries.

- Poor mental health and challenging psychosocial circumstances are also known to commonly affect adolescents.

- This study was done to assess the association between adolescents' mental health and psychosocial circumstances and injury in order to suggest potential points of intervention to improve outcomes.

## What did the researchers do and find?

- We quantified the association between psychosocial circumstances and serious injury occurrence, mechanism, and type in adolescents, taking into account country as well as individual factors that could affect the relationship.

- We found that factors and behaviours such as being bullied and using drugs were strongly associated with an increase in serious injuries, as were indicators of poor mental health such as being too worried to sleep or having attempted suicide.

- We also found that factors that we hypothesised to be protective, such as having close friends or understanding family members or guardians, were not significantly associated with the occurrence of serious injury, apart from in the highest categories, where they were associated with an increase in serious injuries.

- The relationships between psychosocial circumstances and injury were similar across world regions and countries with different income status.

## What do these findings mean?

- All available evidence points to psychosocial factors such as bullying, drug-taking, and poor mental health being strongly associated with occurrence of injury, and this association is consistent across geographical regions and countries with different income status.

- Our findings suggest that multifactorial programmes to target psychosocial factors might reduce the number of serious injuries in adolescents across different countries and contexts.

- In particular, concentrating on reducing bullying and drug use and improving mental health could reduce the number of serious injuries among adolescents in the future.

## Introduction

The World Health Organization (WHO) estimates that 1.2 billion people are adolescents, aged between 10 and 19 years, constituting a sixth of the world's population [1]. Moreover nearly 90% of adolescents live in low- and middle-income countries (LMICs) [1]. In 2016, 1.1 million adolescents, globally, died from preventable or treatable causes. Injury is a major cause of death among adolescents [2], with most of these deaths occurring in LMICs [3]. Even non-fatal serious injuries can cause long-term disabilities and have devastating effects on an individual's overall health and well-being [4]. In particular, a recent study showed that road traffic injuries, self-harm, and interpersonal violence ranked in the top 5 causes of disability-adjusted life years (DALYs) within this age group [5]. With an increasing global population [6], this presents a major public health issue, and it has now been embedded in the Sustainable Development Goals (SDGs). Not only is there a moral imperative to reduce injuries in adolescents, there is also a strong economic argument for doing so [7]. Clearly, to reduce the overall number of preventable deaths and disabilities due to injury and the economic burden on health systems [8,9], it is imperative to address the causes of adolescent injury, especially in LMICs—

where there has been less international focus—in order that targeted and public health policies can be developed and implemented.

In recent years, there has been a shift in attitude and increased attention towards mental health, particularly in adolescents [10]. Adolescence is a critical period in human development, biologically, socially, and emotionally, and recognition of this has accelerated the focus on youth mental health and the potential for preventing poor health in adulthood [11]. There is increasing evidence of associations between mental health or the circumstances in which people live and behaviour that could reduce or increase risks of injury [3,12–35].

The role of psychosocial factors in determining health outcomes as conceptualised by Dahlgren and Whitehead [36] has led to a growing awareness of the importance of and the need to address these factors to improve health, and has widely influenced research [37]. Based on our previous research on resilience in adolescence [38], we developed a framework that organised potential psychosocial factors into 4 broad groups: (1) protective factors indicating a nurturing or supportive environment, (2) risk factors including indicators of risky behaviour, (3) indicators of poor mental health, and (4) contextual factors, encompassing material resources and cultural norms. These multilevel factors are in a dynamic recursive relationship with each other, and, as yet, the relationships between them and injury outcomes are not well understood, particularly in the context of LMICs. To supplement our working knowledge and previous work, we reviewed the literature on unintentional injuries in adolescents [2,11–34] to identify the relevant variables against these different dimensions to develop our analytical framework, and to enable us to hypothesise potential associations, providing avenues for further research on interventions.

The Global School-based Student Health Survey (GSHS) is an ongoing collaborative surveillance project between WHO and the Centers for Disease Control and Prevention [39,40] involving 101 countries and consisting of a series of questions about each adolescent's health and living circumstances. It is a school-based questionnaire with 3 main aims: to encourage countries to prioritise and establish school and youth health programmes and policies, to enable countries to identify the behavioural risk factors and protective factors in key areas among young people in school education (examples of such factors include alcohol and drug use, mental health, and violence and unintentional injury), and to allow international organisations and other nations to be able to make international comparisons on the prevalence of global health behaviours and protective factors [40]. We aimed to ascertain associations between psychosocial factors and the occurrence of any serious injury (intentional or unintentional), the most serious injury type, and the most serious injury mechanism using data from the GSHS, and to compare associations across geographical regions and by level of country development.

## Methods

We conducted a secondary analysis of data collected in the GSHS. This survey, translated to the appropriate local language, is deployed in each country using a 2-stage sampling design. In the first stage, schools are randomly selected from a list of all schools in a country with a probability proportional to enrolment size. In the second stage, classes are randomly selected within the school using systematic equal probability sampling with random start. Surveys are self-completed by students at their schools. The survey consists of a core set of modules asked at all schools and a choice of expanded and country-specific modules. Prior to data collection, the questionnaire was piloted to ensure adequate local comprehension of the survey. Module questions and data are freely available for download and analysis (https://www.cdc.gov/GSHS/ and https://www.who.int/teams/noncommunicable-diseases/surveillance/data).

This study is reported as per the Strengthening the Reporting of Observational Studies in Epidemiology (STROBE) guideline (S1 STROBE Checklist). We limited analysis to data collected between 2009 and 2015 to maximise temporal comparability. Included in the 10 core modules, asked in all countries, are questions on social circumstances, alcohol use, drug use, mental health, physical activity, tobacco use, and violence and unintentional injury. After discussions between the authors, we agreed on variables to be included in the analysis, based upon our knowledge of the literature on the relationships between psychosocial variables and violence or other injuries, and the availability of those variables in the dataset. We documented this set of variables prior to starting the analysis. Questions in the core survey were updated over time, meaning not all countries had data for all variables that the authors considered of potential interest; hence, some variables that were listed in the core set but were found to have limited availability were discarded before we agreed on the final set for inclusion—therefore 'Have you ever had sexual intercourse?' was excluded from the analysis due to limited availability. To be included in this analysis, countries needed to have responses to all questions of relevance.

## Outcomes

Our primary outcome was occurrence of any serious injury (intentional or unintentional), defined as an injury requiring the respondent to miss a day or more of usual activities or needing treatment by a doctor or nurse, in the last 12 months. Participants were asked, 'During the past 12 months, how many times were you seriously injured?' Participants who responded with 1 or more times were categorised as having had a serious injury in the last 12 months.

Our secondary outcomes were (1) the type of injury of the most serious injury that occurred in the last 12 months (not seriously injured; broken bone/dislocated joint; cut or stab wound; concussion/head or neck injury, was knocked out, or could not breathe; gunshot wound; bad burn; poisoned or took too much of a drug; or other) and (2) the mechanism of injury of the most serious injury that occurred in the last 12 months (not seriously injured; motor vehicle accident [MVA]; fall; hit by a falling object; attacked, abused, or fighting; fire or too near a flame or something hot; inhaled or swallowed something bad; or other). The categories of 'other' were captured at the time of the survey by respondents documenting that of 'something else' happened to them. It is not possible to disaggregate this category further.

## Exposure variables

Variables were selected for inclusion based on their availability in the surveys and their hypothesised relationship with the outcome variables. Their inclusion was agreed upon through discussion amongst the authors. For clarity of interpretation of the results, we divided the exposure variables into overarching categories (see Table 1) conceptualised as psychosocial factors—defined as factors that relate to individuals socially, or their thoughts or behaviour. These 'Indicators of psychosocial circumstances' contains markers of risky behaviour, with variables hypothesised to be associated with risk-taking or an increased risk of injury (e.g., smoking, alcohol use, drug use, and physical activity); contextual factors, with variables that could suggest respondents live in an environment that makes them vulnerable and hence more susceptible to injury (e.g., going hungry, being bullied, and feeling lonely) [41]; protective factors, with variables that may suggest a nurturing environment with social support that might lower risk of injury occurrence (e.g., number of friends and having a supportive family); and indicators of poor mental health, with variables previously identified in the General Health Questionnaire for detecting minor psychiatric morbidity (e.g., planned or attempted suicide and being too worried to sleep) [41,42].

**Table 1. Variables and their categories used in the analysis, as captured in the Global School-based Student Health Survey, apart from country-level factors, which are extracted from the World Bank World Development Indicators database and use World Bank classification.**

| Country-level factors | Individual-level factors | | | | | |
|---|---|---|---|---|---|---|
| | Demographic factors | Indicators of psychosocial circumstances | | | | Aggressive behaviour indicators |
| | | Markers of risky behaviour | Contextual factors | Protective factors | Indicators of poor mental health | |
| Income status | Age | Number of days smoked in the past 30 days | Went hungry in the past 30 days | Number of close friends | Considered or planned suicide in the past 12 months[*] | Number of times in a physical fight in the past 12 months |
| High | ≤13 years | Never (0 days) | Never | 0 | No | 0 times |
| Upper middle | 14 or 15 years | Rarely (1 or 2/3–5 days) | Rarely | 1 | Yes | 1 time |
| Lower middle | ≥16 years | Often (6–9/10–19/20–29 days) | Sometimes | 2 | Attempted suicide in the past 12 months | 2 to 3 times |
| Low | Sex | Always (all 30 days) | Most of the time/always | 3 or more | No | ≥4 times |
| World region | Male | Number of days of alcohol use in the past 30 days | Number of days bullied in the past 30 days | Parents or guardians understand problems and worries in the past 30 days | Yes | How many times got into fights as result of alcohol during your lifetime |
| East Asia and the Pacific | Female | Never (0 days) | Never (0 days) | Never | Too worried to sleep in the past 12 months | 0 times |
| South Asia | | Rarely (1 or 2/3–5 days) | Rarely (1 or 2/3–5 days) | Rarely | Never | 1 to 2 times |
| Latin America and the Caribbean | | Often (6–9/10–19/20–29 days) | Often (6–9/10–19/20–29 days) | Sometimes | Rarely | ≥3 times |
| Sub-Saharan Africa | | Always (all 30 days) | Always (all 30 days) | Most of the time/always | Sometimes | Number of times physically attacked in the past 12 months |
| | | Ever used drugs | Felt lonely in the past 12 months | | Most of the time/always | 0 times |
| | | No | Never | | | 1 time |
| | | Yes | Rarely | | | 2 to 3 times |
| | | Physical activity in the past 7 days | Sometimes | | | ≥4 |
| | | 0 days | Most of the time/always | | | |
| | | 1 day | | | | |
| | | 2 days | | | | |
| | | ≥3 days | | | | |

Variable names are presented as shown in the results; the responses that informed each variable are presented in Table A in S1 Text. The original questions are available from https://www.cdc.gov/GSHS/ and https://www.who.int/teams/noncommunicable-diseases/surveillance/systems-tools/global-school-based-student-health-survey).
[*]This variable was formed from 2 survey questions relating to considering/planning suicide (see Table A in S1 Text); a positive response to either question was deemed as being a positive result for this variable.

Control variables included country income status and geographical area (both using World Bank classifications) and aggressive behaviour indicators, which were selected as variables that may suggest that the respondent was more likely to be involved in fights.

Each question in the GSHS allowed a respondent to choose from a number of responses or exposure levels. Where the number of respondents for any exposure level was small, choices were merged (Table A in S1 Text).

## Statistical analysis

None of the variables met criteria for collinearity. After excluding collinearity between variables using generalised variance inflation factors [43,44], logistic regression was used to estimate associations between outcomes and indicators of psychosocial circumstances, controlling for demographic and country-level factors as categorised by the World Bank [45]; the surveys did not contain information on individuals' economic circumstances. We adjusted for study design by using the survey's stratum, weights, and primary sampling units (PSUs). These latter variables were used to account for selection of schools and classrooms, non-responding schools and students, and population distribution by grade and sex, as detailed in the GSHS questionnaire user's guide [46]. The study weights ($W$) were calculated using the equation $W = W1 \times W2 \times f1 \times f2 \times f3$, with the following definitions: W1, the inverse probability of selecting each school; W2, the inverse probability of selecting each classroom; f1 and f2, the non-response adjustment factors at school level and classroom level, respectively; and f3, a post-stratification adjustment factor calculated by sex within grade. The stratum was assigned sequentially starting at schools with the largest enrolment through to schools with the smallest. The PSU was assigned sequentially starting with the classes within the schools with the largest enrolment of students and continuing through those with the smallest enrolment. The weights, stratum, and PSU were provided with the GSHS data. However, to ensure that school assignments were unique between countries, the given stratum assignment was adjusted by adding a 3-digit prefix that specified which country the schools were in.

In subsequent models, we adjusted for aggressive behaviour indicators to see if this adjustment would nullify the effects of other psychosocial factors on injuries. For the primary outcome models, models without country factors as cofounding variables were also fitted, to explore whether country-level factors impacted on associations between psychosocial circumstances and serious injury.

For our primary outcome, we ran logistic regression models and, for our secondary outcomes, multinomial logistic regression models with type of injury and mechanism of injury as the exposure levels. For all models, forced entry was used to allow the effects of variables on the outcome, even if associations were not significant.

Descriptive results are shown as unweighted number and weighted percentage. Results from regression analyses are presented as adjusted odds ratios (ORs) and 95% confidence interval (95% CIs). Results are presented in the text for the odds of the highest compared to lower exposure levels of each variable, unless otherwise stated.

We used R version 3.6.1 (https://www.rstudio.org) to analyse the data, and 2-tailed $p$-values $< 0.05$ were considered statistically significant. Complete case analyses were used.

## Ethical statement

Data collection was approved by each country's local ethics governance committee. Permissions to conduct the survey in each school were obtained from heads of the selected schools and classroom teachers, written or verbal consent to take part in the survey was given by students or their parents, and surveys were completed anonymously (https://www.cdc.gov/GSHS/ and https://www.who.int/teams/noncommunicable-diseases/surveillance/systems-tools/global-school-based-student-health-survey). This study only used pre-existing anonymous survey data, and no further ethical approval was deemed necessary.

## Results

A total of 87,269 participants (46,374 [49.3%] female) living in 26 countries supplied data for the analysis from all World Bank income status levels and 4 geographical regions (Table B in

S1 Text). Most participants were living in lower-middle income countries 81.93%, 15.53% were from upper-middle, 2.2% from low-, and 0.35% from high-income countries. Most participants were from the East Asia and Pacific region (66.83%); South Asia supplied 15.55%, Latin America and the Caribbean 10.00%, and sub-Saharan Africa 7.62% of participants (Table 2).

Of the total population, 36.33% had had a serious injury in the past 12 months (Table 2); the percentage of respondents who had been injured was largest in low-income countries (48.74%). Injuries were most common in sub-Saharan Africa (55.80%). The percentage of respondents who had been injured was similar in each age group. Males experienced a greater number of injuries than females (42.62% of males versus 29.87% of females injured; Table C in S1 Text shows results by sex).

Results from multivariable analysis of associations of indicators of psychosocial circumstances with the primary outcome of any serious injury in the past 12 months, controlling for demographic and country-level factors, are shown in Table 3 and Fig 1. Considering markers of risky behaviour, the relationships between smoking and alcohol use and injury were inconsistent, with some exposures being associated with increased odds of injury, whilst most were not. However, ever having used drugs was significantly associated with injury (OR 2.08, 95% CI 1.73 to 2.49, $p < 0.001$). Physical activity was not associated with an increased odds of injury (OR 1.03, 95% CI 0.92 to 1.14, $p = 0.645$).

The contextual factors, going hungry (OR 1.61, 95% CI 1.30 to 2.01, $p < 0.001$) being bullied (OR 2.45, 95% CI 1.93 to 3.13, $p < 0.001$), and feeling lonely (OR 1.61, 95% CI 1.37 to 1.89, $p < 0.001$), were all significant independent risk factors for a higher odds of injury at all risk exposures, apart from feeling lonely, where the lowest exposure category (rarely felt lonely) was not significant (OR 1.06, 95% CI 0.93 to 1.22, $p = 0.358$).

Considering protective factors, having a greater number of friends and having parents or guardians understanding worries and concerns were not associated with injury odds, apart from a higher odds of injury in participants who reported the greatest number of friends (OR 1.23, 95% CI 1.06 to 1.43, $p = 0.007$) and those who reported having parents being the most understanding (OR 1.13, 95% CI 1.01 to 1.26, $p < 0.036$).

For indicators of poor mental health, attempting suicide in the last 12 months (OR 1.78, CI 1.55 to 2.04, $p < 0.001$) was significantly associated with a greater odds of injury; however, the association for considering suicide was not significant (OR 1.08, CI 0.97 to 1.22, $p = 0.172$). Being too worried to sleep was significantly associated with injury at all levels of exposure (OR 1.80, 95% CI 1.54 to 2.10, $p < 0.001$).

The results of adding the aggressive behaviour indicators to the model with individual- and country-level factors are shown in Table 3 and S1 Fig. All of the aggressive behaviour indicators were significantly associated with an increased odds of injury: being in a physical fight (OR 2.94, 95% CI 2.49 to 3.46, $p < 0.001$), getting into a fight as a result of alcohol (OR 1.44, 95% CI 1.13 to 1.82, $p = 0.003$), and being physically attacked (OR 2.89, 95% CI 2.42 to 3.45, $p < 0.001$). Addition of these variables did not substantially alter the relationships between other variables and the outcome, apart from reducing the association of older age with odds of being injured (OR 0.96, 95% CI 0.84 to 1.07, $p = 0.378$, for ≥16 years versus ≤13 years when aggressive behaviour indicators are added to the model). Adding country-level factors to the full model including aggressive behaviour indicators did not substantially adjust the relationship between serious injury and other exploratory variables (Table D in S1 Text).

Responses to questions on type and mechanism of injury were captured in 87,033 and 84,883 participants, respectively. Table 4 shows the number of responses for each type and mechanism of injury. Tables E–H in S1 Text show characteristics of the populations used in the analyses for type and mechanism of injury.

**Table 2. Characteristics of the total population and by occurrence of any serious injury in the last 12 months.**

| Characteristic | Total | | No serious injuries | | ≥1 serious injury | |
|---|---|---|---|---|---|---|
| | Unweighted *N* | Weighted percent | Unweighted *N* | Weighted percent | Unweighted *N* | Weighted percent |
| **N (Complete Cases)** | 87,269 | 100 | 56,478 | 63.67 | 30,791 | 36.33 |
| **Country-level factors** | | | | | | |
| **Income status** | | | | | | |
| Low | 2,644 | 2.20 | 1,360 | 51.26 | 1,284 | 48.74 |
| Lower middle | 30,007 | 81.93 | 19,099 | 63.86 | 10,908 | 36.14 |
| Upper middle | 46,337 | 15.53 | 30,662 | 64.40 | 15,675 | 35.60 |
| High | 8,281 | 0.35 | 5,357 | 64.02 | 2,924 | 35.98 |
| **World region** | | | | | | |
| East Asia and Pacific | 45,992 | 66.83 | 30,626 | 67.71 | 15,366 | 32.29 |
| South Asia | 2,198 | 15.55 | 1,467 | 57.58 | 731 | 42.42 |
| Latin America and Caribbean | 31,715 | 10.00 | 20,756 | 61.00 | 10,959 | 39.00 |
| Sub-Saharan Africa | 7,364 | 7.62 | 3,629 | 44.20 | 3,735 | 55.80 |
| **Demographic characteristics** | | | | | | |
| **Age** | | | | | | |
| ≤13 years | 20,948 | 30.89 | 13,616 | 65.80 | 7,332 | 34.20 |
| 14 or 15 years | 40,135 | 45.88 | 25,852 | 62.55 | 14,283 | 37.45 |
| ≥16 years | 26,186 | 23.23 | 17,010 | 63.05 | 9,176 | 36.95 |
| **Sex** | | | | | | |
| Male | 40,895 | 50.70 | 23,769 | 57.38 | 17,126 | 42.62 |
| Female | 46,374 | 49.30 | 32,709 | 70.13 | 13,665 | 29.87 |
| **Markers of risky behaviour** | | | | | | |
| **Number of days smoked in the past 30 days** | | | | | | |
| Never (0 days) | 76,958 | 90.30 | 51,294 | 64.97 | 25,664 | 35.03 |
| Rarely (1 or 2/3–5 days) | 6,167 | 6.30 | 3,074 | 47.62 | 3,093 | 52.38 |
| Often (6–9/10–19/20–29 days) | 2,476 | 2.18 | 1,278 | 61.54 | 1,198 | 38.46 |
| Always (all 30 days) | 1,668 | 1.21 | 832 | 54.19 | 836 | 45.81 |
| **Number of days of alcohol use in the past 30 days** | | | | | | |
| Never (0 days) | 67,317 | 89.11 | 45,293 | 65.63 | 22,024 | 34.37 |
| Rarely (1 or 2/3–5 days) | 15,729 | 9.43 | 9,027 | 47.96 | 6,702 | 52.04 |
| Often (6–9/10–19/20–29 days) | 3,664 | 1.25 | 1,891 | 46.31 | 1,773 | 53.69 |
| Always (all 30 days) | 559 | 0.21 | 267 | 39.65 | 292 | 60.35 |
| **Ever used drugs** | | | | | | |
| No | 81,497 | 96.21 | 53,967 | 64.89 | 27,530 | 35.11 |
| Yes | 5,772 | 3.79 | 2,511 | 32.63 | 3,261 | 67.37 |
| **Physical activity in the past 7 days** | | | | | | |
| 0 days | 19,470 | 29.64 | 12,960 | 65.28 | 6,510 | 34.72 |
| 1 day | 17,689 | 22.13 | 11,877 | 64.80 | 5,812 | 35.20 |
| 2 days | 13,049 | 11.29 | 8,831 | 64.47 | 4,218 | 35.53 |
| ≥3 days | 37,061 | 36.94 | 22,810 | 61.46 | 14,251 | 38.54 |
| **Contextual factors** | | | | | | |
| **Went hungry in the past 30 days** | | | | | | |
| Never | 43,236 | 42.16 | 30,558 | 69.92 | 12,678 | 30.08 |
| Rarely | 18,743 | 18.33 | 11,655 | 61.95 | 7,088 | 38.05 |
| Sometimes | 20,764 | 33.34 | 12,074 | 59.27 | 8,690 | 40.73 |
| Most of the time/always | 4,526 | 6.18 | 2,191 | 49.85 | 2,335 | 50.15 |

*(Continued)*

**Table 2.** (Continued)

| Characteristic | Total | | No serious injuries | | ≥1 serious injury | |
|---|---|---|---|---|---|---|
| | Unweighted *N* | Weighted percent | Unweighted *N* | Weighted percent | Unweighted *N* | Weighted percent |
| **Number of days bullied in the past 30 days** | | | | | | |
| Never (0 days) | 65,734 | 72.26 | 46,831 | 71.95 | 18,903 | 28.05 |
| Rarely (1 or 2/3–5 days) | 17,251 | 23.00 | 7,964 | 43.24 | 9,287 | 56.76 |
| Often (6–9/10–19/20–29 days) | 2,853 | 3.45 | 1,089 | 34.51 | 1,764 | 65.49 |
| Always (all 30 days) | 1,431 | 1.29 | 594 | 41.77 | 837 | 58.23 |
| **Felt lonely in the past 12 months** | | | | | | |
| Never | 30,343 | 34.09 | 21,663 | 71.28 | 8,680 | 28.72 |
| Rarely | 21,858 | 19.43 | 14,376 | 67.56 | 7,482 | 32.44 |
| Sometimes | 26,694 | 37.25 | 16,158 | 59.00 | 10,536 | 41.00 |
| Most of the time/always | 8,374 | 9.23 | 4,281 | 46.23 | 4,093 | 53.77 |
| **Protective factors** | | | | | | |
| **Number of close friends** | | | | | | |
| 0 | 4,355 | 4.25 | 2,651 | 58.92 | 1,704 | 41.08 |
| 1 | 8,677 | 10.42 | 5,374 | 61.31 | 3,303 | 38.69 |
| 2 | 11,071 | 12.11 | 6,938 | 61.09 | 4,133 | 38.91 |
| 3 or more | 63,166 | 73.22 | 41,515 | 64.71 | 21,651 | 35.29 |
| **Parents or guardians understand problems and worries** | | | | | | |
| Never | 18,358 | 19.34 | 11,923 | 63.92 | 6,435 | 36.08 |
| Rarely | 14,383 | 12.86 | 9,020 | 59.63 | 5,363 | 40.37 |
| Sometimes | 21,145 | 30.40 | 13,337 | 64.82 | 7,808 | 35.18 |
| Most of the time/always | 33,383 | 37.40 | 22,198 | 63.99 | 11,185 | 36.01 |
| **Indicators of poor mental health** | | | | | | |
| **Considered or planned suicide** | | | | | | |
| No | 73,710 | 88.75 | 49,729 | 65.68 | 23,981 | 34.32 |
| Yes | 13,559 | 11.25 | 6,749 | 47.78 | 6,810 | 52.22 |
| **Attempted suicide** | | | | | | |
| No | 78,359 | 92.42 | 52,566 | 65.75 | 25,793 | 34.25 |
| Yes | 8,910 | 7.58 | 3,912 | 38.25 | 4,998 | 61.75 |
| **Too worried to sleep** | | | | | | |
| Never | 33,966 | 40.04 | 24,713 | 71.70 | 9,253 | 28.30 |
| Rarely | 24,288 | 21.63 | 15,770 | 64.89 | 8,518 | 35.11 |
| Sometimes | 22,817 | 32.23 | 13,146 | 56.72 | 9,671 | 43.28 |
| Most of the time/always | 6,198 | 6.10 | 2,849 | 43.31 | 3,349 | 56.69 |
| **Aggressive behaviour indicators** | | | | | | |
| **Number of times in a physical fight in the past 12 months** | | | | | | |
| 0 times | 63,386 | 74.54 | 45,747 | 71.78 | 17,639 | 28.22 |
| 1 time | 11,279 | 12.96 | 5,789 | 47.38 | 5,490 | 52.62 |
| 2 to 3 times | 7,291 | 7.52 | 3,160 | 35.97 | 4,131 | 64.03 |
| ≥4 times | 5,313 | 4.99 | 1,782 | 26.55 | 3,531 | 73.45 |
| **How many times ever got into fights as a result of alcohol** | | | | | | |
| 0 times | 79,740 | 95.36 | 52,954 | 64.94 | 26,786 | 35.06 |
| 1 to 2 times | 5,315 | 3.30 | 2,637 | 40.79 | 2,678 | 59.21 |
| ≥3 times | 2,214 | 1.34 | 887 | 29.70 | 1,327 | 70.30 |

*(Continued)*

**Table 2.** (Continued)

| Characteristic | Total | | No serious injuries | | ≥1 serious injury | |
|---|---|---|---|---|---|---|
| | Unweighted N | Weighted percent | Unweighted N | Weighted percent | Unweighted N | Weighted percent |
| **Number of times physically attacked in the past 12 months** | | | | | | |
| 0 times | 63,807 | 64.69 | 45,927 | 73.46 | 17,880 | 26.54 |
| 1 time | 9,684 | 13.37 | 4,929 | 53.55 | 4,755 | 46.45 |
| 2 to 3 times | 7,760 | 12.61 | 3,493 | 47.56 | 4,267 | 52.44 |
| ≥4 times | 6,018 | 9.33 | 2,129 | 32.08 | 3,889 | 67.92 |

Results from the multivariable analysis of associations between indicators of psychosocial circumstances and type of injury, controlling for demographic and country-level factors but not adjusting for aggressive behaviour indicators, are shown in Table 5 and Figs 2 and 3. Considering markers of risky behaviour, ever having used drugs was significantly associated with increased odds of broken bone/dislocated joint (OR 1.73, 95% CI 1.37 to 2.18, $p < 0.001$), cut or stab wound (OR 1.73, 95% CI 1.20 to 2.49, $p = 0.004$), and gunshot wound (OR 3.43, 95% CI 1.63 to 7.20, $p < 0.001$). Higher levels of physical activity were significantly associated with broken bone/dislocated joint (OR 1.33, 95% CI 1.11 to 1.60, $p = 0.002$), but there was no consistent relationship seen between other markers of risky behaviour and injury type.

For contextual factors, being bullied was associated with increased odds of all types of injury, but there was not a consistent exposure response. Feeling lonely was associated with a greater odds of broken bone/dislocated joint (OR 1.29, 95% CI 1.02 to 1.65, $p = 0.036$), cut or stab wound (OR 1.44, 95% CI 1.00 to 2.07, $p = 0.050$), head injury (OR 2.33, 95% CI 1.13 to 4.82, $p = 0.022$), and burns (OR 1.75, 95% CI 1.08 to 2.84, $p = 0.023$), but only at higher levels of exposure. Going hungry was not strongly associated with any injury type.

Protective factors were not associated with a lower odds of any type of injury apart from gunshot, where there was a lower odds in respondents with more understanding families or guardians (OR 0.56, 95% CI 0.34 to 0.93, $p = 0.026$).

Considering indicators of poor mental health, attempting suicide was associated with a significantly increased odds of all injury types except burns (OR 1.34, 95% CI 0.84 to 2.15, $p = 0.218$). Being too worried to sleep was associated with an increased odds of a broken bone (OR 1.4, 95% CI 1.17 to 1.68, $p < 0.001$) and head injury (OR 1.90, 95% CI 1.18 to 3.08, $p = 0.009$) at higher exposures, but relationships with other injury types were either non-significant at any exposure (cut or stab wound, gunshot wound, poisoned), or not consistent (bad burn).

Associations between mechanism of injury and indicators of psychosocial circumstances, controlling for demographic and country-level factors but not adjusting for aggressive behaviour indicators, are shown in Table 6. Considering markers of risky behaviour, smoking was associated with a greater odds of MVA at all exposures (OR 1.71, 95% CI 1.14 to 2.55, $p < 0.001$), and with a greater odds of falling and being attacked or abused or fighting but only at low exposure. Alcohol use was inconsistently associated with mechanisms across exposure levels. Ever having used drugs was associated with greater odds of MVA (OR 1.52, 95% CI 1.11 to 2.08, $p = 0.001$), being hit by a falling object (OR 1.88, 95% CI 1.18 to 3.00, $p = 0.008$), and being attacked or abused or fighting (OR 2.11, 95% CI 1.31 to 3.41, $p = 0.002$). Higher levels of physical activity were associated with falls (OR 1.47, 95% CI 1.18 to 1.83, $p < 0.001$, for ≥3 days per week compared with none).

Considering contextual factors, being hungry was associated with falls (OR 1.74, 95% CI 1.29 to 2.35, $p < 0.001$), being hit by a falling object (OR 2.08, 95% CI 1.29 to 3.35, $p = 0.003$),

**Table 3. Multivariable associations between serious injury occurrence, country-level factors, and individual-level factors of risky behaviour, contextual factors, protective factors, and indicators of poor mental health (model 1; model 2 shows the results when aggressive behaviour indicators are added to the model).**

| Variable | Exposure level | Model 1 (unadjusted model) | | | Model 2, controlling for aggressive behaviour indicators | | |
|---|---|---|---|---|---|---|---|
| | | OR | 95% CI | *p*-Value | OR | 95% CI | *p*-Value |
| **Country-level factors** | | | | | | | |
| **Income status** | Low | 0.81 | 0.65–1.01 | 0.056 | 0.89 | 0.71–1.10 | 0.277 |
| | Lower middle (ref) | — | — | — | — | — | — |
| | Upper middle | 1.08 | 1.00–1.16 | 0.042 | 1.06 | 0.99–1.14 | 0.115 |
| | High | 1.21 | 1.07–1.38 | 0.002 | 1.15 | 1.01–1.30 | 0.033 |
| **World region** | East Asia and Pacific (ref) | — | — | — | — | — | — |
| | South Asia | 1.72 | 1.33–2.22 | <0.001 | 1.32 | 1.04–1.67 | 0.023 |
| | Latin America and Caribbean | 1.13 | 1.03–1.24 | 0.007 | 1.15 | 1.05–1.27 | 0.003 |
| | Sub-Saharan Africa | 2.40 | 2.05–2.80 | <0.001 | 2.31 | 1.99–2.68 | <0.001 |
| **Demographic characteristics** | | | | | | | |
| **Age** | ≤13 years (ref) | — | — | — | — | — | — |
| | 14 or 15 years | 0.94 | 0.84–1.06 | 0.316 | 1.04 | 0.93–1.17 | 0.465 |
| | ≥16 years | 0.80 | 0.70–0.91 | 0.001 | 0.95 | 0.84–1.07 | 0.378 |
| **Sex** | Male (ref) | — | — | — | — | — | — |
| | Female | 0.56 | 0.52–0.61 | <0.001 | 0.67 | 0.62–0.72 | <0.001 |
| **Markers of risky behaviour** | | | | | | | |
| **Number of days smoked in the past 30 days** | 0 days (ref) | — | — | — | — | — | — |
| | 1–5 days | 1.23 | 0.94–1.59 | 0.130 | 1.08 | 0.86–1.36 | 0.499 |
| | 6–29 days | 0.65 | 0.47–0.90 | 0.009 | 0.59 | 0.44–0.79 | <0.001 |
| | All 30 days | 0.98 | 0.70–1.36 | 0.881 | 0.72 | 0.51–1.01 | 0.057 |
| **Number of days of alcohol use in the past 30 days** | 0 days (ref) | — | — | — | — | — | — |
| | 1–5 days | 1.43 | 1.28–1.60 | <0.001 | 1.24 | 1.11–1.38 | <0.001 |
| | 6–29 days | 1.19 | 0.96–1.48 | 0.116 | 0.93 | 0.76–1.15 | 0.510 |
| | All 30 days | 1.39 | 0.90–2.14 | 0.138 | 1.07 | 0.69–1.67 | 0.757 |
| **Ever used drugs** | No (ref) | — | — | — | — | — | — |
| | Yes | 2.08 | 1.73–2.49 | <0.001 | 1.80 | 1.49–2.18 | <0.001 |
| **Physical activity in the past 7 days** | 0 days (ref) | — | — | — | — | — | — |
| | 1 day | 1.02 | 0.93–1.11 | 0.726 | 1.01 | 0.92–1.10 | 0.910 |
| | 2 days | 0.97 | 0.86–1.09 | 0.597 | 0.93 | 0.83–1.05 | 0.247 |
| | ≥3 days | 1.03 | 0.92–1.14 | 0.645 | 1.00 | 0.90–1.11 | 0.979 |
| **Contextual factors** | | | | | | | |
| **Went hungry in the past 30 days** | Never (ref) | — | — | — | — | — | — |
| | Rarely | 1.29 | 1.17–1.43 | <0.001 | 1.23 | 1.12–1.35 | <0.001 |
| | Sometimes | 1.32 | 1.19–1.47 | <0.001 | 1.28 | 1.15–1.43 | <0.001 |
| | Most of the time/always | 1.61 | 1.30–2.01 | <0.001 | 1.50 | 1.23–1.84 | <0.001 |
| **Number of days bullied in the past 30 days** | 0 days (ref) | — | — | — | — | — | — |
| | 1–5 days | 2.55 | 2.28–2.85 | <0.001 | 2.03 | 1.84–2.24 | <0.001 |
| | 6–29 days | 3.22 | 2.69–3.84 | <0.001 | 2.24 | 1.83–2.74 | <0.001 |
| | All 30 days | 2.45 | 1.93–3.13 | <0.001 | 1.89 | 1.45–2.46 | <0.001 |
| **Felt lonely in the past 12 months** | Never (ref) | — | — | — | — | — | — |
| | Rarely | 1.06 | 0.93–1.22 | 0.358 | 1.06 | 0.94–1.20 | 0.324 |
| | Sometimes | 1.38 | 1.24–1.54 | <0.001 | 1.30 | 1.16–1.45 | <0.001 |
| | Most of the time/always | 1.61 | 1.37–1.89 | <0.001 | 1.46 | 1.26–1.69 | <0.001 |
| **Protective factors** | | | | | | | |

*(Continued)*

**Table 3.** (Continued)

| Variable | Exposure level | Model 1 (unadjusted model) | | | Model 2, controlling for aggressive behaviour indicators | | |
|---|---|---|---|---|---|---|---|
| | | OR | 95% CI | p-Value | OR | 95% CI | p-Value |
| **Number of close friends** | 0 (ref) | — | — | — | — | — | — |
| | 1 | 1.03 | 0.84–1.25 | 0.790 | 1.06 | 0.87–1.29 | 0.555 |
| | 2 | 1.01 | 0.82–1.23 | 0.964 | 0.97 | 0.79–1.19 | 0.783 |
| | 3 or more | 1.23 | 1.06–1.43 | 0.007 | 1.18 | 1.01–1.37 | 0.041 |
| **Parents or guardians understand problems and worries** | Never (ref) | — | — | — | — | — | — |
| | Rarely | 1.10 | 0.93–1.30 | 0.283 | 1.09 | 0.93–1.28 | 0.268 |
| | Sometimes | 0.99 | 0.89–1.10 | 0.820 | 1.01 | 0.91–1.12 | 0.911 |
| | Most of the time/always | 1.13 | 1.01–1.26 | 0.036 | 1.14 | 1.02–1.27 | 0.023 |
| **Indicators of poor mental health** | | | | | | | |
| **Considered or planned suicide** | No (ref) | — | — | — | — | — | — |
| | Yes | 1.08 | 0.97–1.22 | 0.172 | 1.00 | 0.89–1.12 | 0.987 |
| **Attempted suicide** | No (ref) | — | — | — | — | — | — |
| | Yes | 1.78 | 1.55–2.04 | <0.001 | 1.65 | 1.42–1.91 | <0.001 |
| **Too worried to sleep** | Never (ref) | — | — | — | — | — | — |
| | Rarely | 1.17 | 1.05–1.30 | 0.004 | 1.15 | 1.04–1.27 | 0.005 |
| | Sometimes | 1.50 | 1.36–1.64 | <0.001 | 1.41 | 1.27–1.55 | <0.001 |
| | Most of the time/always | 1.80 | 1.54–2.10 | <0.001 | 1.60 | 1.37–1.88 | <0.001 |
| **Aggressive behaviour indicators** | | | | | | | |
| **Number of times in a physical fight** | 0 times (ref) | — | — | — | — | — | — |
| | 1 time | — | — | — | 1.81 | 1.60–2.04 | <0.001 |
| | 2–3 times | — | — | — | 2.27 | 2.01–2.56 | <0.001 |
| | ≥4 times | — | — | — | 2.94 | 2.49–3.46 | <0.001 |
| **How many times got into fights as a result of alcohol** | 0 times (ref) | — | — | — | — | — | — |
| | 1–2 times | — | — | — | 1.27 | 1.08–1.50 | 0.003 |
| | ≥3 times | — | — | — | 1.44 | 1.13–1.82 | 0.003 |
| **Number of times physically attacked** | 0 times (ref) | — | — | — | — | — | — |
| | 1 time | — | — | — | 1.59 | 1.40–1.81 | <0.001 |
| | 2–3 times | — | — | — | 1.95 | 1.71–2.24 | <0.001 |
| | ≥4 times | — | — | — | 2.89 | 2.42–3.45 | <0.001 |

For both models, adjustment for non-response was done as detailed in Methods. OR, odds ratio.

being attacked (OR 1.57, 95% CI 1.04 to 2.38, $p = 0.031$, for the middle level of exposure; significance was lost at the highest level), or inhaling or swallowing something bad (OR 1.88, 95% CI 1.13 to 3.13, $p = 0.016$), but there was no association at any level of exposure of hunger with fire (OR 1.67, 95% CI 0.88 to 3.18, $p = 0.117$). Being bullied was associated with all mechanisms at all levels of exposure, apart from fire, where significance was lost in the highest exposure category (OR 2.60, 95% CI 0.52 to 12.92, $p = 0.243$), and inhaling or swallowing something bad, where it was lost in the middle exposure category (OR 1.69, 95% CI 0.77 to 3.72, $p = 0.192$); feeling lonely at higher exposures was associated with being hit by a falling object (OR 1.68, 95% CI 1.01 to 2.81, $p = 0.047$), being attacked (OR 2.11, 95% CI 1.32 to 3.38, $p = 0.002$), and inhaling or swallowing something bad (OR 1.60, 95% CI 1.04 to 2.49, $p = 0.034$).

Having a larger number of friends had no protective effect for any mechanism; however, having increased levels of family understanding was associated with an increased odds of falling (OR 1.36, 95% CI 1.02 to 1.82, $p = 0.038$).

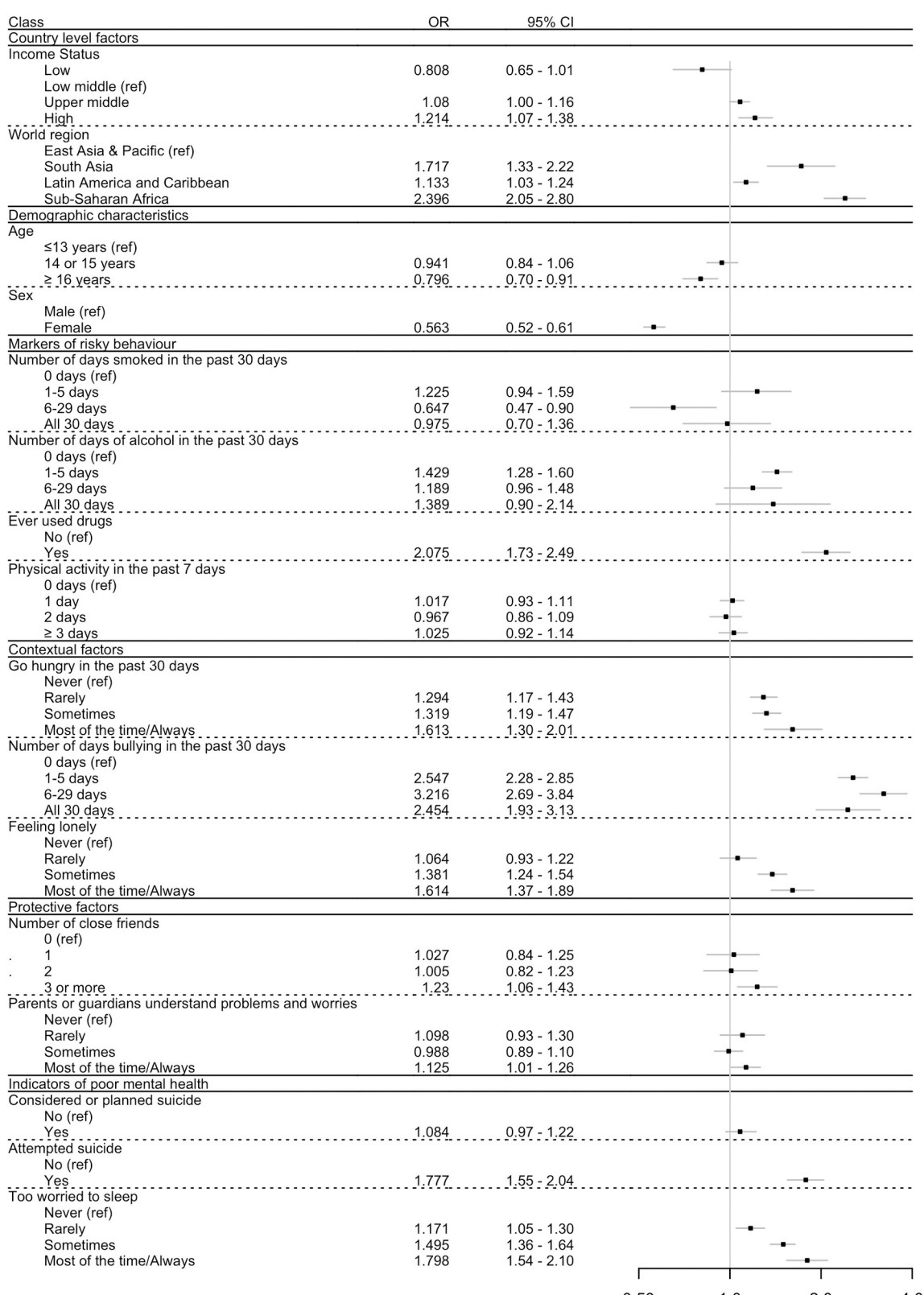

| Class | OR | 95% CI |
|---|---|---|
| **Country level factors** | | |
| Income Status | | |
| Low | 0.808 | 0.65 - 1.01 |
| Low middle (ref) | | |
| Upper middle | 1.08 | 1.00 - 1.16 |
| High | 1.214 | 1.07 - 1.38 |
| World region | | |
| East Asia & Pacific (ref) | | |
| South Asia | 1.717 | 1.33 - 2.22 |
| Latin America and Caribbean | 1.133 | 1.03 - 1.24 |
| Sub-Saharan Africa | 2.396 | 2.05 - 2.80 |
| **Demographic characteristics** | | |
| Age | | |
| ≤13 years (ref) | | |
| 14 or 15 years | 0.941 | 0.84 - 1.06 |
| ≥ 16 years | 0.796 | 0.70 - 0.91 |
| Sex | | |
| Male (ref) | | |
| Female | 0.563 | 0.52 - 0.61 |
| **Markers of risky behaviour** | | |
| Number of days smoked in the past 30 days | | |
| 0 days (ref) | | |
| 1-5 days | 1.225 | 0.94 - 1.59 |
| 6-29 days | 0.647 | 0.47 - 0.90 |
| All 30 days | 0.975 | 0.70 - 1.36 |
| Number of days of alcohol in the past 30 days | | |
| 0 days (ref) | | |
| 1-5 days | 1.429 | 1.28 - 1.60 |
| 6-29 days | 1.189 | 0.96 - 1.48 |
| All 30 days | 1.389 | 0.90 - 2.14 |
| Ever used drugs | | |
| No (ref) | | |
| Yes | 2.075 | 1.73 - 2.49 |
| Physical activity in the past 7 days | | |
| 0 days (ref) | | |
| 1 day | 1.017 | 0.93 - 1.11 |
| 2 days | 0.967 | 0.86 - 1.09 |
| ≥ 3 days | 1.025 | 0.92 - 1.14 |
| **Contextual factors** | | |
| Go hungry in the past 30 days | | |
| Never (ref) | | |
| Rarely | 1.294 | 1.17 - 1.43 |
| Sometimes | 1.319 | 1.19 - 1.47 |
| Most of the time/Always | 1.613 | 1.30 - 2.01 |
| Number of days bullying in the past 30 days | | |
| 0 days (ref) | | |
| 1-5 days | 2.547 | 2.28 - 2.85 |
| 6-29 days | 3.216 | 2.69 - 3.84 |
| All 30 days | 2.454 | 1.93 - 3.13 |
| Feeling lonely | | |
| Never (ref) | | |
| Rarely | 1.064 | 0.93 - 1.22 |
| Sometimes | 1.381 | 1.24 - 1.54 |
| Most of the time/Always | 1.614 | 1.37 - 1.89 |
| **Protective factors** | | |
| Number of close friends | | |
| 0 (ref) | | |
| 1 | 1.027 | 0.84 - 1.25 |
| 2 | 1.005 | 0.82 - 1.23 |
| 3 or more | 1.23 | 1.06 - 1.43 |
| Parents or guardians understand problems and worries | | |
| Never (ref) | | |
| Rarely | 1.098 | 0.93 - 1.30 |
| Sometimes | 0.988 | 0.89 - 1.10 |
| Most of the time/Always | 1.125 | 1.01 - 1.26 |
| **Indicators of poor mental health** | | |
| Considered or planned suicide | | |
| No (ref) | | |
| Yes | 1.084 | 0.97 - 1.22 |
| Attempted suicide | | |
| No (ref) | | |
| Yes | 1.777 | 1.55 - 2.04 |
| Too worried to sleep | | |
| Never (ref) | | |
| Rarely | 1.171 | 1.05 - 1.30 |
| Sometimes | 1.495 | 1.36 - 1.64 |
| Most of the time/Always | 1.798 | 1.54 - 2.10 |

0.50    1.0    2.0    4.0

**Fig 1. Forest plot showing the multivariable associations (as odds ratio [OR] and 95% confidence interval [95% CI]) between serious injury occurrence and psychosocial circumstances, adjusting for country-level factors and demographic characteristics,**

**but not for aggressive behaviour indicators.** Adjustment for non-response rate of participants was done as described in Methods. *p*-Values can be found in Table 3 (Model 1).

For indicators of poor mental health, having previously attempted suicide was associated with an increased odds of MVA (OR 1.79, 95% CI 1.15 to 2.80, *p* = 0.010), fall (OR 1.28, 95% CI 1.02 to 1.61, *p* = 0.032), being hit by a falling object (OR 1.72, 95% CI 1.23 to 2.42, *p* = 0.002), and being attacked (OR 1.47, 95% CI 1.00 to 2.16, *p* = 0.047). However, considering suicide was only associated with an increased odds of being attacked (OR 1.50, 95% CI 1.02 to 2.19, *p* = 0.038). A higher level of exposure of being too worried to sleep was associated with an increased odds of all injury mechanisms.

## Discussion

In this multi-country analysis, there was a high prevalence of self-reported serious injuries, with the greatest proportion of these occurring in low-income countries and sub-Saharan Africa. We found that serious injuries were strongly associated with multiple factors across the categories of markers of risky behaviour, contextual factors, protective factors, indicators of poor mental health, and aggressive behaviour indicators. Controlling for geographic region and income status of countries did not change the results, suggesting that these relationships may be consistent across world regions. Associations with injury type and mechanism varied.

We found, after controlling for demographic and country-level factors, that indicators of psychosocial circumstances associated with risky behaviour, being in a vulnerable environment, and having poor mental health were positively associated with the occurrence of serious injury. Others have found some associations between individual factors in these categories and

**Table 4. Number of serious injuries by type and mechanism (numbers of cases used for this analysis and their characteristics are shown in Tables E–H in S1 Text).**

| Variable | Total | |
|---|---|---|
| | **Unweighted *N*** | **Weighted percent** |
| **During the past 12 months, what was the most serious injury that happened to you?** | | |
| *N* (complete cases by injury type) | 87,033 | 100 |
| Not seriously injured | 63,993 | 72.29 |
| Broken bone/dislocated joint | 5,212 | 7.16 |
| Cut or stab wound | 4,231 | 6.16 |
| Concussion/head or neck injury, was knocked out, or could not breathe | 1,949 | 2.48 |
| Gunshot wound | 278 | 0.23 |
| Bad burn | 798 | 1.08 |
| Poisoned or took too much of a drug | 258 | 0.23 |
| Something else happened | 10,314 | 10.38 |
| **During the past 12 months, what was the major cause of the most serious injury that happened to you?** | | |
| *N* (complete cases by injury mechanism) | 84,883 | 100 |
| Not seriously injured | 62,232 | 74.23 |
| Motor vehicle accident | 2,725 | 4.11 |
| Fall | 8,243 | 10.17 |
| Hit by a falling object | 2,167 | 2.65 |
| Attacked, abused, or fighting | 1,346 | 1.66 |
| Fire or too near a flame or something hot | 469 | 0.53 |
| Inhaled or swallowed something bad | 496 | 0.72 |
| Something else | 7,205 | 5.92 |

**Table 5. Multivariable associations between serious injury type, country-level factors, and individual-level factors of risky behaviour, contextual factors, protective factors, and indicators of poor mental health.**

| Variable | Exposure level | Broken bone or dislocated joint | | | Cut or stab wound | | | Concussion/head or neck injury, was knocked out, or could not breathe | | | Gunshot wound | | | Bad burn | | | Poisoned or took too much of a drug | | | Something else happened | | |
|---|---|---|---|---|---|---|---|---|---|---|---|---|---|---|---|---|---|---|---|---|---|---|
| | | OR | 95% CI | p-Value | OR | 95% CI | p-Value | OR | 95% CI | p-Value | OR | 95% CI | p-Value | OR | 95% CI | p-Value | OR | 95% CI | p-Value | OR | 95% CI | p-Value |
| **Country-level factors** | | | | | | | | | | | | | | | | | | | | | | |
| Income status | Low | 0.70 | 0.49–1.00 | 0.048 | 0.69 | 0.51–0.94 | 0.018 | 0.57 | 0.28–1.13 | 0.108 | 2.00 | 0.55–7.32 | 0.293 | 0.71 | 0.30–1.70 | 0.439 | 0.05 | 0.00–25660.19 | 0.646 | 0.52 | 0.34–0.80 | 0.003 |
| | Lower middle (ref) | — | — | — | — | — | — | — | — | — | — | — | — | — | — | — | — | — | — | — | — | — |
| | Upper middle | 0.73 | 0.66–0.82 | <0.001 | 2.62 | 1.71–4.02 | <0.001 | 1.25 | 0.93–1.69 | 0.133 | 2.02 | 0.90–4.54 | 0.090 | 0.66 | 0.44–0.97 | 0.035 | 1.33 | 0.50–3.56 | 0.570 | 1.09 | 0.94–1.26 | 0.251 |
| | High | 1.08 | 0.74–1.58 | 0.705 | 6.19 | 1.47–26.03 | 0.013 | 1.52 | 0.03–72.20 | 0.831 | 7.97 | 0.19–336.86 | 0.277 | 1.01 | 0.06–17.38 | 0.993 | 9.57 | 0.19–470.95 | 0.256 | 1.13 | 0.82–1.55 | 0.448 |
| World region | East Asia and Pacific (ref) | — | — | — | — | — | — | — | — | — | — | — | — | — | — | — | — | — | — | — | — | — |
| | South Asia | 0.63 | 0.41–0.95 | 0.026 | 13.88 | 6.14–31.39 | <0.001 | 3.24 | 1.61–6.55 | 0.001 | 1.72 | 0.75–3.94 | 0.197 | 1.57 | 0.74–3.33 | 0.244 | 2.68 | 1.02–7.08 | 0.046 | 0.87 | 0.61–1.23 | 0.431 |
| | Latin America and Caribbean | 0.99 | 0.84–1.17 | 0.935 | 0.55 | 0.41–0.75 | <0.001 | 0.94 | 0.73–1.22 | 0.653 | 0.57 | 0.11–2.96 | 0.503 | 1.25 | 0.81–1.95 | 0.314 | 0.51 | 0.06–4.18 | 0.528 | 2.14 | 1.79–2.56 | <0.001 |
| | Sub-Saharan Africa | 1.34 | 1.03–1.73 | 0.028 | 6.26 | 3.48–11.25 | <0.001 | 1.70 | 0.99–2.91 | 0.054 | 1.81 | 0.69–4.73 | 0.228 | 2.04 | 1.25–3.32 | 0.004 | 2.42 | 0.87–6.77 | 0.091 | 1.22 | 0.94–1.56 | 0.132 |
| **Demographic characteristics** | | | | | | | | | | | | | | | | | | | | | | |
| Age | ≤13 years (ref) | — | — | — | — | — | — | — | — | — | — | — | — | — | — | — | — | — | — | — | — | — |
| | 14 or 15 years | 1.04 | 0.91–1.18 | 0.589 | 0.70 | 0.52–0.94 | 0.019 | 0.78 | 0.55–1.11 | 0.164 | 0.47 | 0.31–0.72 | <0.001 | 0.89 | 0.62–1.28 | 0.525 | 0.58 | 0.33–1.01 | 0.054 | 0.95 | 0.79–1.14 | 0.563 |
| | ≥16 years | 0.84 | 0.73–0.98 | 0.024 | 0.65 | 0.49–0.87 | 0.004 | 0.72 | 0.51–1.01 | 0.055 | 0.63 | 0.37–1.07 | 0.090 | 0.77 | 0.53–1.10 | 0.152 | 0.96 | 0.42–2.18 | 0.917 | 0.92 | 0.76–1.11 | 0.369 |
| Sex | Male (ref) | — | — | — | — | — | — | — | — | — | — | — | — | — | — | — | — | — | — | — | — | — |
| | Female | 0.44 | 0.36–0.56 | <0.001 | 0.52 | 0.41–0.67 | <0.001 | 0.73 | 0.57–0.94 | 0.014 | 0.57 | 0.39–0.85 | 0.005 | 0.63 | 0.44–0.89 | 0.008 | 0.85 | 0.51–1.42 | 0.545 | 0.63 | 0.53–0.74 | <0.001 |
| **Markers of risky behaviour** | | | | | | | | | | | | | | | | | | | | | | |
| Number of days smoked in the past 30 days | 0 days (ref) | — | — | — | — | — | — | — | — | — | — | — | — | — | — | — | — | — | — | — | — | — |
| | 1–5 days | 1.53 | 1.27–1.83 | <0.001 | 0.84 | 0.50–1.41 | 0.501 | 1.40 | 0.89–2.20 | 0.146 | 0.95 | 0.40–2.28 | 0.912 | 0.74 | 0.43–1.28 | 0.281 | 0.73 | 0.28–1.85 | 0.502 | 1.18 | 0.96–1.44 | 0.115 |
| | 6–29 days | 0.90 | 0.65–1.23 | 0.490 | 0.62 | 0.28–1.34 | 0.224 | 0.68 | 0.43–1.08 | 0.100 | 0.68 | 0.10–4.51 | 0.686 | 0.88 | 0.43–1.83 | 0.733 | 1.52 | 0.19–12.53 | 0.697 | 0.70 | 0.49–0.99 | 0.044 |
| | All 30 days | 0.86 | 0.58–1.28 | 0.453 | 0.96 | 0.50–1.86 | 0.907 | 1.28 | 0.78–2.11 | 0.335 | 0.51 | 0.01–24.23 | 0.732 | 0.19 | 0.00–8039.8 | 0.760 | 3.58 | 0.04–319.76 | 0.578 | 1.05 | 0.70–1.58 | 0.825 |
| Number of days of alcohol use in the past 30 days | 0 days (ref) | — | — | — | — | — | — | — | — | — | — | — | — | — | — | — | — | — | — | — | — | — |
| | 1–5 days | 1.18 | 0.99–1.41 | 0.062 | 1.58 | 1.21–2.05 | <0.001 | 1.43 | 1.09–1.87 | 0.011 | 1.09 | 0.56–2.11 | 0.802 | 1.54 | 1.09–2.17 | 0.013 | 1.81 | 0.88–3.72 | 0.110 | 1.24 | 1.03–1.50 | 0.022 |
| | 6–29 days | 1.10 | 0.75–1.60 | 0.621 | 1.21 | 0.59–2.50 | 0.603 | 1.05 | 0.62–1.77 | 0.853 | 1.97 | 0.36–10.77 | 0.436 | 1.67 | 0.22–12.77 | 0.619 | 1.43 | 0.13–15.70 | 0.770 | 1.18 | 0.93–1.49 | 0.186 |
| | All 30 days | 1.33 | 0.74–2.41 | 0.343 | 1.19 | 0.24–5.82 | 0.829 | 0.31 | $0.00–8.686 \times 10^5$ | 0.878 | 1.66 | 0.02–163.30 | 0.828 | 0.37 | $0.00–0.643 \times 10^{11}$ | 0.942 | 0.00 | $0.00–1.369 \times 10^{13}$ | 0.529 | 1.00 | 0.45–2.22 | 0.997 |
| Ever used drugs | No (ref) | — | — | — | — | — | — | — | — | — | — | — | — | — | — | — | — | — | — | — | — | — |
| | Yes | 1.73 | 1.37–2.18 | <0.001 | 1.73 | 1.20–2.49 | 0.004 | 1.37 | 0.90–2.10 | 0.143 | 3.43 | 1.63–7.20 | 0.001 | 1.88 | 0.95–3.73 | 0.072 | 2.48 | 0.82–7.48 | 0.106 | 1.27 | 1.02–1.58 | 0.034 |
| Physical activity in the past 7 days | 0 days (ref) | — | — | — | — | — | — | — | — | — | — | — | — | — | — | — | — | — | — | — | — | — |
| | 1 day | 1.25 | 1.06–1.48 | 0.010 | 1.22 | 0.91–1.65 | 0.185 | 0.86 | 0.67–1.10 | 0.234 | 0.90 | 0.54–1.50 | 0.693 | 0.93 | 0.63–1.38 | 0.726 | 0.78 | 0.47–1.31 | 0.348 | 1.08 | 0.89–1.31 | 0.427 |
| | 2 days | 1.15 | 0.95–1.37 | 0.146 | 1.20 | 0.86–1.67 | 0.278 | 0.97 | 0.74–1.27 | 0.825 | 1.03 | 0.54–1.94 | 0.935 | 0.67 | 0.45–1.00 | 0.052 | 1.64 | 0.98–2.75 | 0.059 | 1.13 | 0.92–1.39 | 0.237 |
| | ≥3 days | 1.33 | 1.11–1.60 | 0.002 | 1.03 | 0.82–1.29 | 0.801 | 1.04 | 0.77–1.40 | 0.813 | 0.90 | 0.54–1.48 | 0.673 | 0.72 | 0.51–1.01 | 0.054 | 0.69 | 0.44–1.06 | 0.091 | 1.33 | 1.07–1.65 | <0.006 |
| **Contextual factors** | | | | | | | | | | | | | | | | | | | | | | |

(Continued)

**Table 5.** (Continued)

| Variable | Exposure level | Broken bone or dislocated joint | | | Cut or stab wound | | | Concussion/head or neck injury, was knocked out, or could not breathe | | | Gunshot wound | | | Bad burn | | | Poisoned or took too much of a drug | | | Something else happened | | |
|---|---|---|---|---|---|---|---|---|---|---|---|---|---|---|---|---|---|---|---|---|---|---|
| | | OR | 95% CI | p-Value | OR | 95% CI | p-Value | OR | 95% CI | p-Value | OR | 95% CI | p-Value | OR | 95% CI | p-Value | OR | 95% CI | p-Value | OR | 95% CI | p-Value |
| Went hungry in the past 30 days | Never (ref) | — | — | — | — | — | — | — | — | — | — | — | — | — | — | — | — | — | — | — | — | — |
| | Rarely | 1.08 | 0.91–1.29 | 0.357 | 1.37 | 0.84–2.23 | 0.214 | 1.28 | 0.91–1.82 | 0.162 | 0.90 | 0.54–1.50 | 0.691 | 0.94 | 0.69–1.29 | 0.703 | 0.66 | 0.36–1.21 | 0.179 | 1.21 | 1.05–1.39 | 0.010 |
| | Sometimes | 1.15 | 0.96–1.38 | 0.126 | 1.38 | 1.04–1.82 | 0.025 | 1.24 | 0.88–1.76 | 0.216 | 0.81 | 0.51–1.30 | 0.383 | 1.15 | 0.74–1.79 | 0.534 | 0.55 | 0.36–0.86 | 0.008 | 1.30 | 1.08–1.56 | 0.006 |
| | Most of the time/always | 1.30 | 0.99–1.71 | 0.058 | 1.45 | 0.97–2.17 | 0.070 | 1.43 | 0.94–2.19 | 0.098 | 1.22 | 0.64–2.34 | 0.544 | 1.54 | 0.88–2.70 | 0.135 | 0.78 | 0.37–1.64 | 0.515 | 1.40 | 1.08–1.81 | 0.010 |
| Number of days bullied in the past 30 days | 0 days (ref) | — | — | — | — | — | — | — | — | — | — | — | — | — | — | — | — | — | — | — | — | — |
| | 1–5 days | 1.81 | 1.47–2.24 | <0.001 | 2.49 | 1.57–3.94 | <0.001 | 2.49 | 1.39–4.46 | 0.002 | 2.13 | 1.25–3.60 | 0.005 | 2.60 | 1.41–4.82 | 0.002 | 2.11 | 1.15–3.86 | 0.016 | 2.05 | 1.61–2.61 | <0.001 |
| | 6–29 days | 1.94 | 1.50–2.50 | <0.001 | 2.67 | 1.38–5.16 | 0.004 | 2.66 | 1.60–4.43 | <0.001 | 3.30 | 0.99–11.02 | 0.052 | 4.50 | 1.65–12.26 | 0.003 | 2.70 | 0.73–10.00 | 0.137 | 2.61 | 1.94–3.49 | <0.001 |
| | All 30 days | 1.52 | 1.13–2.04 | 0.005 | 1.53 | 0.82–2.85 | 0.180 | 2.37 | 1.11–5.06 | 0.026 | 8.33 | 1.74–39.96 | 0.008 | 2.10 | 0.87–5.05 | 0.099 | 3.98 | 0.71–22.22 | 0.115 | 1.96 | 1.36–2.81 | <0.001 |
| Felt lonely in the past 12 months | Never (ref) | — | — | — | — | — | — | — | — | — | — | — | — | — | — | — | — | — | — | — | — | — |
| | Rarely | 1.13 | 0.91–1.41 | 0.279 | 0.80 | 0.57–1.11 | 0.176 | 1.06 | 0.71–1.59 | 0.763 | 0.75 | 0.46–1.22 | 0.240 | 0.93 | 0.58–1.47 | 0.746 | 0.73 | 0.40–1.30 | 0.283 | 0.95 | 0.75–1.19 | 0.638 |
| | Sometimes | 1.37 | 1.12–1.68 | 0.002 | 1.20 | 0.90–1.59 | 0.224 | 1.30 | 0.83–2.04 | 0.254 | 0.71 | 0.46–1.10 | 0.129 | 1.07 | 0.66–1.75 | 0.782 | 0.80 | 0.45–1.42 | 0.441 | 1.26 | 1.00–1.60 | 0.052 |
| | Most of the time/always | 1.29 | 1.02–1.65 | 0.036 | 1.44 | 1.00–2.07 | 0.050 | 2.33 | 1.13–4.82 | 0.022 | 0.79 | 0.42–1.47 | 0.452 | 1.75 | 1.08–2.84 | 0.023 | 1.37 | 0.64–2.94 | 0.425 | 1.75 | 1.25–2.45 | 0.001 |
| **Protective factors** | | | | | | | | | | | | | | | | | | | | | | |
| Number of close friends | 0 (ref) | — | — | — | — | — | — | — | — | — | — | — | — | — | — | — | — | — | — | — | — | — |
| | 1 | 0.99 | 0.73–1.35 | 0.943 | 0.95 | 0.66–1.36 | 0.777 | 0.77 | 0.35–1.66 | 0.499 | 1.13 | 0.01–154.04 | 0.962 | 1.17 | 0.49–2.83 | 0.720 | 1.66 | 0.01–368.86 | 0.855 | 0.81 | 0.55–1.18 | 0.262 |
| | 2 | 0.89 | 0.63–1.26 | 0.497 | 1.00 | 0.71–1.40 | 0.996 | 0.87 | 0.40–1.87 | 0.716 | 1.11 | 0.01–136.96 | 0.965 | 0.80 | 0.27–2.35 | 0.686 | 1.67 | 0.01–319.13 | 0.849 | 0.77 | 0.55–1.09 | 0.143 |
| | 3 or more | 0.95 | 0.66–1.38 | 0.806 | 1.11 | 0.69–1.78 | 0.661 | 0.91 | 0.46–1.77 | 0.771 | 0.77 | 0.01–101.49 | 0.917 | 0.72 | 0.26–1.98 | 0.522 | 1.64 | 0.01–242.00 | 0.846 | 1.03 | 0.72–1.47 | 0.891 |
| Parents or guardians understand problems and worries | Never (ref) | — | — | — | — | — | — | — | — | — | — | — | — | — | — | — | — | — | — | — | — | — |
| | Rarely | 1.03 | 0.85–1.23 | 0.794 | 0.87 | 0.62–1.22 | 0.427 | 0.92 | 0.63–1.36 | 0.685 | 0.87 | 0.55–1.37 | 0.551 | 0.98 | 0.55–1.72 | 0.933 | 1.01 | 0.59–1.71 | 0.980 | 1.14 | 0.90–1.44 | 0.268 |
| | Sometimes | 1.07 | 0.88–1.30 | 0.489 | 0.85 | 0.62–1.17 | 0.319 | 0.81 | 0.55–1.17 | 0.262 | 0.69 | 0.42–1.13 | 0.142 | 1.18 | 0.75–1.83 | 0.475 | 0.97 | 0.52–1.81 | 0.913 | 0.98 | 0.79–1.22 | 0.851 |
| | Most of the time/always | 1.11 | 0.91–1.37 | 0.310 | 0.93 | 0.68–1.29 | 0.675 | 0.82 | 0.60–1.13 | 0.222 | 0.56 | 0.34–0.93 | 0.026 | 1.09 | 0.70–1.70 | 0.695 | 1.13 | 0.64–1.98 | 0.678 | 1.06 | 0.84–1.33 | 0.633 |
| **Indicators of poor mental health** | | | | | | | | | | | | | | | | | | | | | | |
| Considered or planned suicide | No (ref) | — | — | — | — | — | — | — | — | — | — | — | — | — | — | — | — | — | — | — | — | — |
| | Yes | 1.15 | 0.98–1.34 | 0.081 | 1.10 | 0.81–1.49 | 0.533 | 1.12 | 0.79–1.58 | 0.532 | 1.25 | 0.78–2.00 | 0.348 | 0.89 | 0.55–1.44 | 0.637 | 1.28 | 0.76–2.15 | 0.364 | 1.05 | 0.89–1.24 | 0.590 |
| Attempted suicide | No (ref) | — | — | — | — | — | — | — | — | — | — | — | — | — | — | — | — | — | — | — | — | — |
| | Yes | 1.33 | 1.07–1.64 | 0.009 | 1.60 | 1.12–2.27 | 0.009 | 2.20 | 1.49–3.25 | <0.001 | 1.60 | 1.04–2.47 | 0.034 | 1.34 | 0.84–2.15 | 0.218 | 2.17 | 1.16–4.06 | 0.016 | 1.26 | 1.05–1.51 | 0.012 |
| Too worried to sleep | Never (ref) | — | — | — | — | — | — | — | — | — | — | — | — | — | — | — | — | — | — | — | — | — |
| | Rarely | 1.01 | 0.84–1.21 | 0.947 | 0.96 | 0.69–1.32 | 0.778 | 1.00 | 0.79–1.27 | 0.987 | 0.73 | 0.40–1.34 | 0.312 | 1.46 | 0.80–2.67 | 0.223 | 0.63 | 0.37–1.09 | 0.099 | 1.35 | 1.04–1.74 | 0.023 |
| | Sometimes | 1.41 | 1.15–1.74 | 0.001 | 1.12 | 0.89–1.41 | 0.335 | 1.53 | 1.05–2.22 | 0.026 | 1.28 | 0.68–2.44 | 0.446 | 2.30 | 1.03–5.15 | 0.043 | 1.10 | 0.63–1.92 | 0.730 | 1.82 | 1.44–2.30 | <0.001 |
| | Most of the time/always | 1.40 | 1.17–1.68 | <0.001 | 1.03 | 0.76–1.40 | 0.843 | 1.90 | 1.18–3.08 | 0.009 | 1.73 | 0.77–3.90 | 0.184 | 1.78 | 0.87–3.63 | 0.116 | 1.16 | 0.51–2.66 | 0.723 | 2.03 | 1.54–2.67 | <0.001 |

The model is not adjusted for aggressive behaviour indicators. Adjustment for non-response was done as detailed in Methods. OR, odds ratio.

**Table 6. Multivariable associations between injury mechanism, country-level factors, and individual-level factors of risky behaviour, contextual factors, protective factors, and indicators of poor mental health.**

| | | Motor vehicle accident or hit by a motor vehicle | | | Fell | | | Something fell on me or hit me | | | Attacked or abused or fighting with someone | | | Fire or too near a flame or something hot | | | Inhaled or swallowed something bad for me | | | Something else caused my injury | | |
|---|---|---|---|---|---|---|---|---|---|---|---|---|---|---|---|---|---|---|---|---|---|---|---|
| | | OR | 95% CI | p-Value | OR | 95% CI | p-Value | OR | 95% CI | p-Value | OR | 95% CI | p-Value | OR | 95% CI | p-Value | OR | 95% CI | p-Value | OR | 95% CI | p-Value |
| **Country-level factors** | | | | | | | | | | | | | | | | | | | | | | |
| Income status | Low | 1.04 | 0.69–1.56 | 0.852 | 0.49 | 0.35–0.69 | <0.001 | 0.17 | 0.03–1.21 | 0.077 | 0.85 | 0.35–2.07 | 0.722 | 0.45 | 0.04–5.18 | 0.522 | 1.71 | 0.72–4.05 | 0.222 | 0.20 | 0.09–0.42 | <0.001 |
| | Lower middle (ref) | — | — | — | — | — | — | — | — | — | — | — | — | — | — | — | — | — | — | — | — | — |
| | Upper middle | 1.09 | 0.85–1.40 | 0.490 | 1.08 | 0.93–1.25 | 0.339 | 1.10 | 0.90–1.34 | 0.365 | 0.96 | 0.73–1.27 | 0.776 | 0.83 | 0.54–1.28 | 0.404 | 0.83 | 0.52–1.31 | 0.411 | 1.44 | 1.12–1.85 | 0.005 |
| | High | 0.00 | 0.00–3.2765 × 10$^{18}$ | 0.719 | 0.79 | 0.43–1.43 | 0.430 | 0.91 | 0.01–115.91 | 0.970 | 0.11 | 0.00–540.08 | 0.610 | 0.39 | 0.00–216.20 | 0.768 | 0.58 | 0.02–18.15 | 0.757 | 1.66 | 0.93–2.96 | 0.088 |
| **World region** | East Asia and Pacific (ref) | — | — | — | — | — | — | — | — | — | — | — | — | — | — | — | — | — | — | — | — | — |
| | Latin America and Caribbean | 0.39 | 0.24–0.64 | <0.001 | 1.74 | 1.41–2.15 | <0.001 | 0.93 | 0.69–1.25 | 0.639 | 0.91 | 0.60–1.39 | 0.668 | 1.21 | 0.60–2.45 | 0.601 | 0.78 | 0.38–1.60 | 0.505 | 2.01 | 1.61–2.50 | <0.001 |
| | Sub-Saharan Africa | 1.02 | 0.71–1.46 | 0.921 | 1.79 | 1.33–2.42 | <0.001 | 3.89 | 2.10–7.19 | <0.001 | 2.20 | 1.29–3.74 | 0.004 | 4.34 | 1.96–9.62 | <0.001 | 2.50 | 1.28–4.89 | 0.008 | 2.79 | 1.74–4.48 | <0.001 |
| **Demographic characteristics** | | | | | | | | | | | | | | | | | | | | | | |
| Age | ≤13 years (ref) | — | — | — | — | — | — | — | — | — | — | — | — | — | — | — | — | — | — | — | — | — |
| | 14 or 15 years | 1.36 | 0.91–2.04 | 0.139 | 0.84 | 0.72–0.98 | 0.028 | 1.11 | 0.78–1.57 | 0.569 | 1.11 | 0.78–1.57 | 0.574 | 0.93 | 0.60–1.46 | 0.761 | 0.85 | 0.56–1.30 | 0.457 | 1.06 | 0.80–1.40 | 0.696 |
| | ≥16 years | 1.33 | 0.93–1.89 | 0.119 | 0.61 | 0.50–0.74 | <0.001 | 0.79 | 0.62–0.99 | 0.044 | 0.78 | 0.56–1.07 | 0.125 | 0.64 | 0.40–1.00 | 0.051 | 0.78 | 0.55–1.13 | 0.189 | 1.09 | 0.86–1.37 | 0.491 |
| Sex | Male (ref) | — | — | — | — | — | — | — | — | — | — | — | — | — | — | — | — | — | — | — | — | — |
| | Female | 0.51 | 0.36–0.72 | <0.001 | 0.64 | 0.55–0.73 | <0.001 | 0.63 | 0.50–0.79 | <0.001 | 0.48 | 0.37–0.62 | <0.001 | 0.94 | 0.67–1.32 | 0.726 | 0.83 | 0.60–1.15 | 0.260 | 0.63 | 0.52–0.76 | <0.001 |
| **Markers of risky behaviour** | | | | | | | | | | | | | | | | | | | | | | |
| Number of days smoked in the past 30 days | 0 days (ref) | — | — | — | — | — | — | — | — | — | — | — | — | — | — | — | — | — | — | — | — | — |
| | 1–5 days | 2.40 | 1.64–3.51 | <0.001 | 1.54 | 1.22–1.95 | <0.001 | 1.09 | 0.78–1.52 | 0.634 | 2.43 | 1.48–3.99 | <0.001 | 2.20 | 0.97–5.03 | 0.060 | 1.20 | 0.66–2.16 | 0.555 | 1.24 | 0.98–1.58 | 0.078 |
| | 6–29 days | 1.94 | 1.28–2.94 | 0.002 | 1.15 | 0.75–1.77 | 0.517 | 1.14 | 0.68–1.91 | 0.630 | 1.53 | 0.86–2.71 | 0.149 | 1.83 | 0.44–7.60 | 0.404 | 1.97 | 0.68–5.69 | 0.209 | 0.88 | 0.55–1.40 | 0.579 |
| | All 30 days | 1.71 | 1.14–2.55 | 0.009 | 0.95 | 0.64–1.39 | 0.773 | 0.74 | 0.36–1.50 | 0.400 | 2.34 | 1.14–4.81 | 0.021 | 0.88 | 0.02–36.06 | 0.947 | 0.62 | 0.05–8.08 | 0.715 | 0.98 | 0.50–1.93 | 0.950 |
| Number of days of alcohol use in the past 30 days | 0 days (ref) | — | — | — | — | — | — | — | — | — | — | — | — | — | — | — | — | — | — | — | — | — |
| | 1–5 days | 1.39 | 1.07–1.81 | 0.015 | 0.99 | 0.81–1.22 | 0.916 | 1.35 | 0.95–1.92 | 0.098 | 2.26 | 1.35–3.80 | 0.002 | 1.14 | 0.65–1.99 | 0.646 | 1.53 | 1.00–2.34 | 0.052 | 1.27 | 1.01–1.59 | 0.022 |
| | 6–29 days | 1.19 | 0.71–1.97 | 0.513 | 0.89 | 0.64–1.25 | 0.493 | 1.60 | 0.77–3.32 | 0.205 | 2.12 | 1.23–3.68 | 0.007 | 0.88 | 0.00–332.47 | 0.966 | 1.66 | 0.52–5.26 | 0.393 | 1.33 | 0.95–1.86 | 0.186 |
| | All 30 days | 1.33 | 0.13–13.35 | 0.807 | 0.66 | 0.01–36.03 | 0.838 | 1.63 | 0.14–19.10 | 0.698 | 1.26 | 0.02–79.95 | 0.913 | 0.00 | 0.00–8.220 × 10$^{29}$ | 0.685 | 0.00 | 0.00–4.636 × 10$^{29}$ | 0.748 | 1.45 | 0.22–9.43 | 0.997 |
| Ever used drugs | No (ref) | — | — | — | — | — | — | — | — | — | — | — | — | — | — | — | — | — | — | — | — | — |
| | Yes | 1.52 | 1.11–2.08 | 0.010 | 1.20 | 0.82–1.74 | 0.348 | 1.88 | 1.18–3.00 | 0.008 | 2.11 | 1.31–3.41 | 0.002 | 1.64 | 0.79–3.39 | 0.184 | 1.14 | 0.56–2.32 | 0.723 | 1.18 | 0.83–1.68 | 0.367 |
| Physical activity in the past 7 days | 0 days (ref) | — | — | — | — | — | — | — | — | — | — | — | — | — | — | — | — | — | — | — | — | — |
| | 1 day | 1.25 | 0.94–1.66 | 0.132 | 1.24 | 1.00–1.53 | 0.049 | 1.14 | 0.88–1.48 | 0.319 | 1.10 | 0.82–1.48 | 0.516 | 1.00 | 0.61–1.66 | 0.988 | 1.08 | 0.70–1.68 | 0.722 | 1.28 | 0.88–1.86 | 0.193 |
| | 2 days | 1.14 | 0.86–1.52 | 0.356 | 1.20 | 0.98–1.48 | 0.077 | 1.04 | 0.71–1.52 | 0.842 | 0.80 | 0.51–1.27 | 0.348 | 1.02 | 0.56–1.85 | 0.949 | 1.06 | 0.66–1.72 | 0.803 | 1.28 | 1.00–1.65 | 0.055 |
| | ≥3 days | 1.41 | 0.95–2.09 | 0.088 | 1.47 | 1.18–1.83 | 0.001 | 1.32 | 0.89–1.95 | 0.171 | 0.96 | 0.68–1.37 | 0.830 | 1.05 | 0.66–1.67 | 0.829 | 1.15 | 0.79–1.68 | 0.476 | 1.91 | 1.42–2.56 | <0.001 |
| **Contextual factors** | | | | | | | | | | | | | | | | | | | | | | |
| Went hungry in the past 30 days | Never (ref) | — | — | — | — | — | — | — | — | — | — | — | — | — | — | — | — | — | — | — | — | — |
| | Rarely | 0.96 | 0.75–1.21 | 0.702 | 1.17 | 0.91–1.49 | 0.229 | 1.31 | 0.92–1.86 | 0.135 | 1.28 | 0.86–1.91 | 0.228 | 0.87 | 0.58–1.29 | 0.476 | 0.93 | 0.59–1.48 | 0.768 | 1.46 | 1.21–1.77 | <0.001 |
| | Sometimes | 1.17 | 0.97–1.42 | 0.107 | 1.42 | 1.18–1.72 | <0.001 | 1.46 | 1.03–2.08 | 0.034 | 1.57 | 1.04–2.38 | 0.031 | 1.36 | 0.94–1.98 | 0.107 | 1.21 | 0.85–1.73 | 0.280 | 1.24 | 0.95–1.63 | 0.109 |
| | Most of the time/always | 1.16 | 0.85–1.56 | 0.350 | 1.74 | 1.29–2.35 | <0.001 | 2.08 | 1.29–3.35 | 0.003 | 1.66 | 0.99–2.80 | 0.056 | 1.67 | 0.88–3.18 | 0.117 | 1.88 | 1.13–3.13 | 0.016 | 1.27 | 0.91–1.78 | 0.157 |
| Number of days bullied in the past 30 days | 0 days (ref) | — | — | — | — | — | — | — | — | — | — | — | — | — | — | — | — | — | — | — | — | — |
| | 1–5 days | 2.00 | 1.35–2.94 | <0.001 | 2.25 | 1.72–2.95 | <0.001 | 2.71 | 1.43–5.13 | 0.002 | 2.57 | 1.47–4.48 | 0.001 | 1.70 | 1.16–2.49 | 0.007 | 2.41 | 1.69–3.43 | <0.001 | 2.10 | 1.56–2.82 | <0.001 |
| | 6–29 days | 1.85 | 1.28–2.68 | 0.001 | 2.21 | 1.55–3.13 | <0.001 | 3.87 | 2.06–7.29 | <0.001 | 2.71 | 1.38–5.32 | 0.004 | 2.75 | 1.25–6.07 | 0.012 | 1.69 | 0.77–3.72 | 0.192 | 2.41 | 1.62–3.59 | <0.001 |
| | All 30 days | 2.02 | 1.14–3.56 | 0.016 | 1.76 | 1.10–2.83 | 0.019 | 2.27 | 1.09–4.74 | 0.029 | 3.84 | 1.35–10.90 | 0.012 | 2.60 | 0.52–12.92 | 0.243 | 3.64 | 1.23–10.80 | 0.020 | 2.41 | 1.59–3.67 | <0.001 |

*(Continued)*

**Table 6.** (Continued)

| | | Motor vehicle accident or hit by a motor vehicle | | | Fell | | | Something fell on me or hit me | | | Attacked or abused or fighting with someone | | | Fire or too near a flame or something hot | | | Inhaled or swallowed something bad for me | | | Something else caused my injury | | |
|---|---|---|---|---|---|---|---|---|---|---|---|---|---|---|---|---|---|---|---|---|---|---|---|---|
| | | OR | 95% CI | p-Value | OR | 95% CI | p-Value | OR | 95% CI | p-Value | OR | 95% CI | p-Value | OR | 95% CI | p-Value | OR | 95% CI | p-Value | OR | 95% CI | p-Value |
| **Felt lonely in the past 12 months** | Never (ref) | — | — | — | — | — | — | — | — | — | — | — | — | — | — | — | — | — | — | — | — | — |
| | Rarely | 1.02 | 0.80–1.31 | 0.845 | 0.97 | 0.78–1.21 | 0.767 | 1.06 | 0.74–1.50 | 0.760 | 1.17 | 0.74–1.86 | 0.496 | 1.44 | 0.81–2.56 | 0.214 | 0.87 | 0.59–1.29 | 0.492 | 1.27 | 0.95–1.71 | 0.112 |
| | Sometimes | 1.12 | 0.87–1.45 | 0.374 | 1.07 | 0.89–1.29 | 0.448 | 1.42 | 0.91–2.22 | 0.120 | 1.46 | 0.90–2.39 | 0.129 | 1.33 | 0.77–2.29 | 0.309 | 1.26 | 0.85–1.87 | 0.257 | 1.41 | 0.98–2.04 | 0.063 |
| | Most of the time/always | 1.13 | 0.82–1.56 | 0.446 | 1.06 | 0.87–1.29 | 0.568 | 1.68 | 1.01–2.81 | 0.047 | 2.11 | 1.32–3.38 | 0.002 | 1.09 | 0.48–2.48 | 0.830 | 1.60 | 1.04–2.49 | 0.034 | 2.20 | 1.46–3.31 | <0.001 |
| **Protective factors** | | | | | | | | | | | | | | | | | | | | | | |
| **Number of close friends** | 0 (ref) | — | — | — | — | — | — | — | — | — | — | — | — | — | — | — | — | — | — | — | — | — |
| | 1 | 1.35 | 0.76–2.39 | 0.312 | 0.94 | 0.67–1.32 | 0.705 | 1.10 | 0.60–2.04 | 0.757 | 1.30 | 0.37–4.65 | 0.684 | 1.41 | 0.04–49.69 | 0.850 | 1.54 | 0.06–42.31 | 0.797 | 0.91 | 0.48–1.72 | 0.780 |
| | 2 | 1.12 | 0.62–1.99 | 0.713 | 0.87 | 0.61–1.24 | 0.438 | 1.03 | 0.56–1.90 | 0.919 | 0.86 | 0.25–3.03 | 0.816 | 1.43 | 0.03–60.81 | 0.851 | 1.25 | 0.05–32.80 | 0.893 | 0.90 | 0.52–1.53 | 0.683 |
| | 3 or more | 1.02 | 0.55–1.92 | 0.947 | 0.99 | 0.72–1.36 | 0.961 | 1.17 | 0.59–2.33 | 0.656 | 1.31 | 0.45–3.82 | 0.620 | 1.04 | 0.03–37.93 | 0.982 | 1.34 | 0.05–34.76 | 0.859 | 1.13 | 0.62–2.08 | 0.686 |
| **Parents or guardians understand problems and worries** | Never (ref) | — | — | — | — | — | — | — | — | — | — | — | — | — | — | — | — | — | — | — | — | — |
| | Rarely | 1.13 | 0.85–1.49 | 0.393 | 1.16 | 0.92–1.48 | 0.215 | 0.94 | 0.70–1.26 | 0.683 | 0.84 | 0.61–1.17 | 0.306 | 1.17 | 0.62–2.19 | 0.628 | 0.93 | 0.55–1.57 | 0.778 | 1.16 | 0.93–1.44 | 0.187 |
| | Sometimes | 1.20 | 0.89–1.62 | 0.226 | 1.32 | 1.07–1.64 | 0.011 | 0.98 | 0.76–1.27 | 0.874 | 0.87 | 0.58–1.31 | 0.512 | 1.08 | 0.56–2.08 | 0.815 | 1.27 | 0.76–2.12 | 0.354 | 1.17 | 0.96–1.42 | 0.127 |
| | Most of the time/always | 1.21 | 0.93–1.56 | 0.153 | 1.36 | 1.02–1.82 | 0.038 | 1.05 | 0.77–1.44 | 0.744 | 0.86 | 0.59–1.24 | 0.414 | 1.01 | 0.52–1.99 | 0.968 | 1.15 | 0.69–1.92 | 0.603 | 1.16 | 0.86–1.56 | 0.324 |
| **Indicators of poor mental health** | | | | | | | | | | | | | | | | | | | | | | |
| **Considered or planned suicide** | No (ref) | — | — | — | — | — | — | — | — | — | — | — | — | — | — | — | — | — | — | — | — | — |
| | Yes | 1.32 | 0.92–1.87 | 0.128 | 1.15 | 0.95–1.40 | 0.144 | 1.15 | 0.86–1.54 | 0.347 | 1.50 | 1.02–2.19 | 0.038 | 1.08 | 0.73–1.60 | 0.697 | 1.44 | 0.98–2.11 | 0.066 | 1.06 | 0.81–1.38 | 0.668 |
| **Attempted suicide** | No (ref) | — | — | — | — | — | — | — | — | — | — | — | — | — | — | — | — | — | — | — | — | — |
| | Yes | 1.79 | 1.15–2.80 | 0.010 | 1.28 | 1.02–1.61 | 0.032 | 1.72 | 1.23–2.42 | 0.002 | 1.47 | 1.00–2.16 | 0.047 | 1.31 | 0.87–2.00 | 0.200 | 1.29 | 0.88–1.91 | 0.197 | 1.25 | 1.00–1.56 | 0.051 |
| **Too worried to sleep** | Never (ref) | — | — | — | — | — | — | — | — | — | — | — | — | — | — | — | — | — | — | — | — | — |
| | Rarely | 1.21 | 0.86–1.68 | 0.273 | 1.17 | 0.92–1.50 | 0.199 | 1.53 | 0.82–2.85 | 0.179 | 1.91 | 1.02–3.60 | 0.044 | 0.85 | 0.54–1.35 | 0.496 | 1.41 | 0.89–2.23 | 0.141 | 1.54 | 1.14–2.09 | 0.005 |
| | Sometimes | 1.56 | 1.11–2.20 | 0.010 | 1.68 | 1.37–2.06 | <0.001 | 1.74 | 0.95–3.20 | 0.075 | 1.92 | 0.98–3.76 | 0.059 | 1.43 | 0.93–2.19 | 0.100 | 1.52 | 1.05–2.20 | 0.027 | 1.52 | 1.09–2.13 | 0.014 |
| | Most of the time/always | 1.80 | 1.20–2.69 | 0.004 | 1.98 | 1.37–2.86 | <0.001 | 2.18 | 1.12–4.28 | 0.023 | 3.33 | 1.57–7.09 | 0.002 | 2.23 | 1.10–4.52 | 0.027 | 1.84 | 1.06–3.21 | 0.031 | 2.02 | 1.37–2.95 | <0.001 |

The model does not adjust for aggressive behaviour indicators. Adjustment for non-response was done as detailed in Methods. OR, odds ratio.

injury in smaller or single-country studies [12,14–16]. However, our more extensive analysis, including a larger number of variables and countries, allows for deeper understanding of the relationships. That we found a relationship between these variables that is not substantially nor significantly adjusted by adding country income status or geographical area to the model suggests that improving the psychosocial environment for adolescents across multiple country settings could have benefits for prevention of injury.

Interestingly, factors that we initially hypothesised would be associated with a more supportive psychosocial environment—for example, having a greater number of friends or more understanding parents or guardians—were not associated with greater injury risk. Although this was initially surprising and went against our hypothesis, in some settings friends and family could be supportive of criminogenic behaviour, leading to increased exposure to injuries [47]. Thus, in totality, our findings suggest that for injury reduction, the immediate focus of actions needs to be on reducing negative psychosocial inputs—reducing risky behaviour, hunger, bullying, and loneliness, and improving mental health—and that improving presumed protective factors of having a supportive family or guardian may be less beneficial, although of undoubted positive benefit for mental health. Different types and mechanisms of injury were

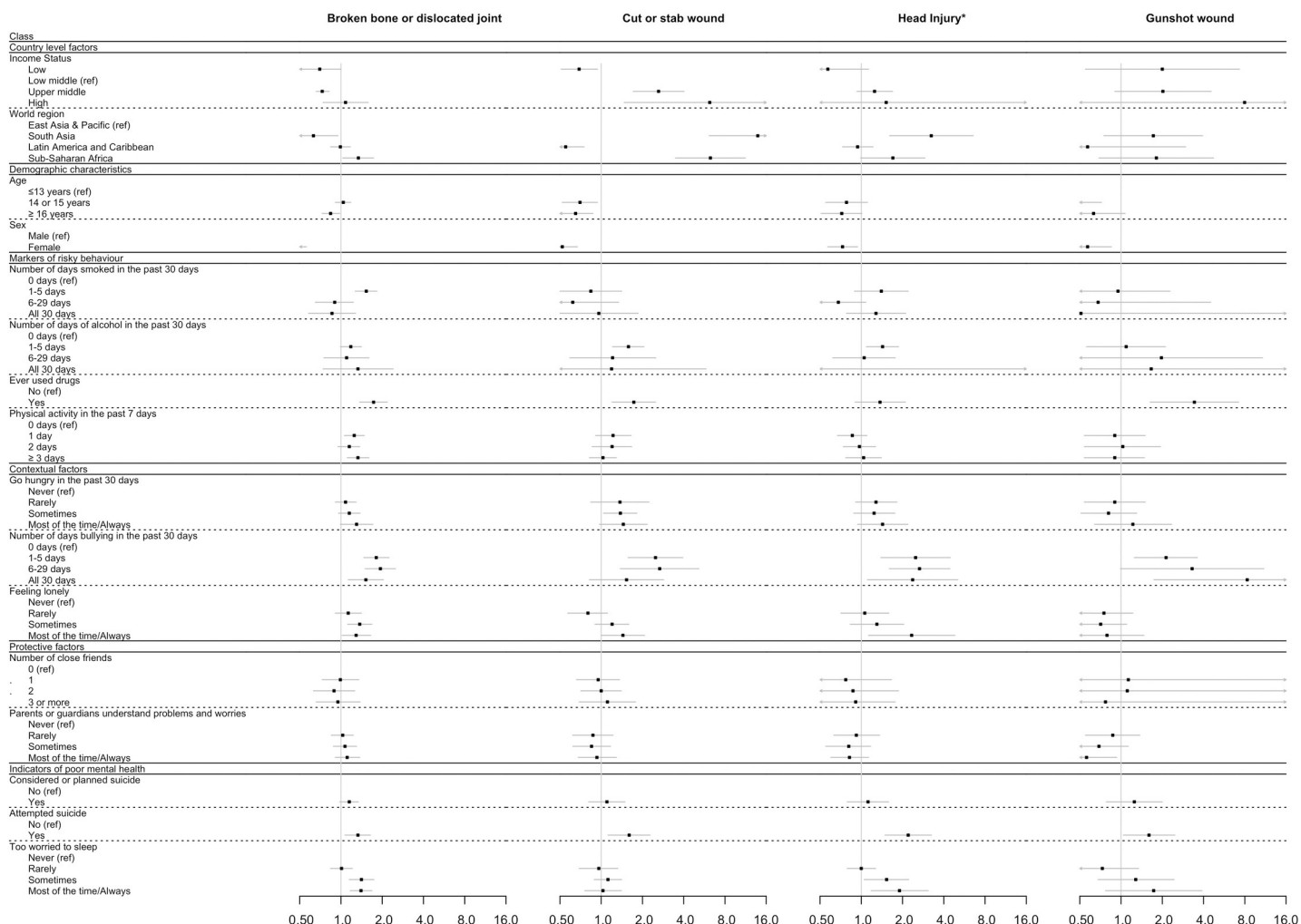

**Fig 2. Forest plot showing the multivariable association (as odds ratio [OR] and 95% confidence interval [95% CI]) between serious injury type (broken bone or dislocated joint, cut or stab wound, head injury, and gunshot wound) and markers of psychosocial circumstances, adjusting for country-level factors and demographic characteristics, but not for aggressive behaviour indicators.** Adjustment for non-response rate of participants was done as described in Methods. *p*-Values can be found in Table 5. *Head Injury refers to the variable 'concussion or other head or neck injury, was knocked out, or could not breathe'.

variably associated with psychosocial factors. This is as expected given that various psychosocial environments may lead to different injury patterns, and it is likely that actions to improve the psychosocial well-being of adolescents may reduce some mechanisms and types of injuries more substantially than others, as is supported by other evidence [19–24].

Of all the psychosocial factors, the marker of risky behaviour, drug use, and the contextual factor, bullying, were most strongly associated with the occurrence of any serious injury, showing these as immediate targets for intervention. A recent systematic review identified that bullying was associated with adverse health and psychosocial outcomes [25], including in academic achievement and social functioning. The strongest, and probably causal, association in that review was between bullying and a range of mental health problems, including anxiety, depression, self-harm, and suicidal ideation [25]. However, in our model, bullying remained a strong independent predictive factor of odds of injury even when controlling for variables indicating poor mental health, suggesting that it is not solely acting through these variables. Moreover, bullying could lead to social exclusion and drive risk-taking behaviour—and thence

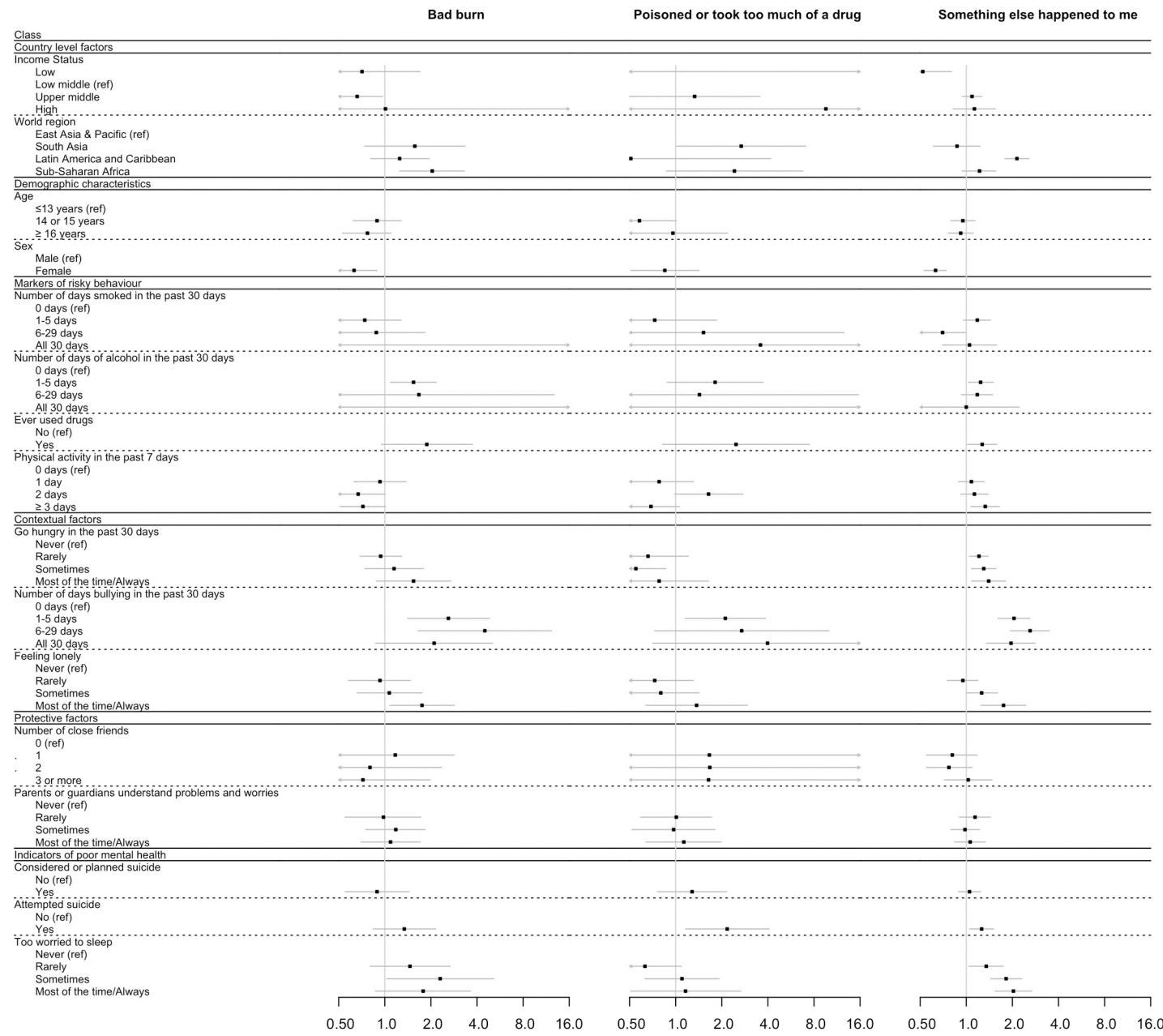

**Fig 3. Forest plot showing the multivariable association (as odds ratio [OR] and 95% confidence interval [95% CI]) between serious injury type (bad burn, poisoned or took too much of a drug, and something else happened) and markers of psychosocial circumstances, adjusting for country-level factors and demographic characteristics, but not for aggressive behaviour indicators.** Adjustment for non-response rate of participants was done as described in Methods. *p*-Values can be found in Table 5.

injuries—as a compensation mechanism to overcome the stress of bullying and social exclusion [48–50]. It is possible that serious injuries could cause bullying, as visible injuries make children appear vulnerable, which could consequently lead to them being bullied. However, there is little evidence for this, and it is likely, as others have found, that low perceived social status leads to bullying, which leads directly to injury via violent acts [24,26,27]. That gunshot

wound was the type of injury most strongly associated with bullying and being attacked or abused or fighting gives weight to this hypothesis.

Moreover, drug use was significantly associated with the injury types broken bone/dislocated joint, cut or stab wound, and gunshot wound, and with the mechanisms of being hit by a falling object and being attacked or abused or fighting. Again, these findings would be expected as these are the types and mechanisms of injuries that are associated with the violence that is often seen with drug-taking and gang-related behaviour. Still, it is possible that the injuries could be the cause of pain leading to drug use, but the mechanisms of injury that were associated with this risky behaviour make this directionality unlikely [28–30]. Other markers of risky behaviour, alcohol use and smoking, were less consistently associated with a serious injury occurrence. However, getting into trouble as a result of alcohol use was associated with increased odds of injury—perhaps because of associated physical violence, as suggested by the association between alcohol and the injury mechanism of being attacked or abused or fighting, as has been seen in other studies [32–34].

We included physical activity as a marker of physical risky behaviour, given our hypothesis that physical activity can increase the risk of injuries via, for example, falls [51–53]. However, in our study, when controlling for covariables, physical activity was not significantly associated with serious injury occurrence. Nevertheless, physical activity at higher levels became a significant risk factor for broken bone/dislocated joint, which could be due to falls during individual exercise or injuries sustained in contact sports. The increased odds of injury were, however, low, and only seen in the highest exposure group. Potential reasons for this include that physical activity at its extreme level could lead to overexertion, which is a leading cause of injuries, particularly in older children, with injuries being more common in high-intensity sports, or higher levels of physical activity increasing the exposure time in which injuries can happen [54]. Given that physical activity is an essential mediator of good health in later life and habits from childhood are often carried forward to later life, this finding shouldn't be used to discourage physical activity in adolescents.

In addition to the associations between bullying and serious injury, the contextual factors going hungry and feeling lonely also were strongly and independently associated with the occurrence of any serious injury. Going hungry was not associated with any particular injury type; however, it was associated with a number of mechanisms. Feeling lonely was associated with a number of injury types and mechanisms but mostly at higher exposures. It is unlikely that being hungry or lonely directly caused injuries or that being injured led to going hungry, apart from in some extreme situations where children are prevented from securing food by their injury. But these factors may reflect an environment that doesn't promote resilience—a trait that is associated with a lower risk of injury [17,18]. These factors also may be associated with a context that is likely to make children vulnerable to injuries, as has been shown for poverty [55]. Also, feeling lonely could be an indicator of poor mental status, increasing the risk of suicide and other risky behaviour.

Interestingly, factors that could be assumed to be associated with a positive psychosocial environment and thus be protective against injuries—having more friends and having understanding parents or guardians—were associated with increased odds of serious injury in our analysis, whereas others have suggested that these factors may promote resilience and protect from injuries and violence [24,56]. We are unable to explore the reasons behind our findings in this cross-sectional study, but it may be that having a larger number of friends was associated with increased peer pressure to undertake risky behaviour or with being involved in large gangs [56], with related violence [28,29]. That said, our findings do not suggest that particular mechanisms or types of injury associated with gang violence are more common in adolescents with more friends. Considering the protective factor of family understanding, it may be that

parents or guardians are more sympathetic after their offspring have been injured; this could explain the positive association between parental understanding and the mechanism of falls. However, for gunshot wounds, the relationship is in the expected direction: Having an understanding family is associated lower odds of the outcome. Although we have found an association between family understanding and number of friends and injuries that is independent of sex, others have found differences by sex. For example, an Israeli study found that risk-taking behaviour by adolescent males was related to orientation towards peers and peer pressure, whilst for females the relationship with parents was the predominant factor in risk behaviour —with more supportive relationships leading to less risk-taking [56].

Markers of poor mental health status, namely having attempted suicide or being too worried to sleep, were associated with a higher odds of injury. For attempting suicide, the associations were significant across most types and mechanisms of injury. It may be that being injured, or resultant disability from injury, led to suicide attempts; indeed, an association between injury occurrence in adolescents and subsequent poor mental health has been found previously [57]. However, the relationship between injuries and poor mental health has been shown to be reciprocal, with poor mental health also associated with greater future occurrence of injuries. From our cross-sectional data, it is not possible to ascertain the directionality of the relationship, but if poor mental health was mainly a result of injury, we would expect the association to be seen across the entirety of injury mechanisms and types, without the exceptions that were seen. Therefore, it is likely that the association that we found between markers of poor mental health and injury is explained by the association that others have found between poor mental health and greater risk-taking behaviour and substance abuse [58], although the literature is limited to settings in high-income countries. It also may be that there is a common origin, and poor mental health is reflective of a broader negative context and socioeconomic disadvantage in which adolescents who also suffer injuries live. That we found that attempting suicide, but not considering or planning suicide, was associated with increased odds of injury was interesting and not due to there being too few people who had considered suicide to detect an effect. This deserves further exploration in future studies. Additionally, feeling lonely could be considered an indicator of poor mental health and was also associated with a higher risk of injuries.

Being physically attacked and in a physical fight were both unsurprisingly associated with a higher odds of injury. However, from adding these variables in separately through a sensitivity analysis, we see that they did not moderate the relationship between injury odds and other psychosocial variables. They did, however, nullify the association between the older age groups and injury occurrence, perhaps due to those in older age groups being more likely to engage in aggressive behaviours.

This study shows the scope for intervention to prevent and reduce the incidence of injuries in adolescence, which requires focus on the psychosocial context of young people's lives. In particular, it indicates the need for holistic interventions to reduce bullying and substance and alcohol use, alongside improved access to preventative measures (e.g., to reduce loneliness) and access to mental health support in school settings and the community. It also highlights the need for more fine-grained studies, using qualitative and ethnographic methods, to understand the nature of the associations reported here, and the mechanisms underpinning them.

The study has several limitations, the main of which is that the cross-sectional nature means that it is not possible to determine the directionality of the relationships between injuries and independent variables. Previous investigators in smaller studies have found that some of the exploratory variables tested in our study are predeterminants of injury [14–16,26,59–61]. That said, the potential for injury being a contributor to poor psychosocial indicators remains, and further work is needed to determine the directionality of these associations. Also,

we recognise that our definition of psychosocial factors is not necessarily the only way of categorising these variables. For example, we categorised feeling lonely along with other contextual variables, considering it as being indicative of the individual's social status. However, we do acknowledge that feeling lonely may indicate a state of mind rather than an actuality, and hence could be considered a marker of poor mental health. Nevertheless, the categorisation of variables does not influence their relationship with the outcome variables.

Additionally, the nature of the survey methodology meant that respondents were all in school, which itself may be protective against injuries and poor mental health [35]. A broader household survey that captured responses from adolescents who were not in school may have found stronger associations between psychosocial factors and injuries. The lack of harmony of questions across countries and time frames means not all countries in which the questionnaire was administered were included in this analysis, and in particular, there were no countries from Europe and Central Asia, or the Middle East and North Africa, and only small numbers of respondents were included from high-income countries. The survey was completed by self-report, without strict definitions given for each question, therefore leaving room for students to interpret the questions in ways meaningful to them, which also may differ by cultural contexts, even though these were controlled for to a certain extent by adding country geographical region to the model. The time frames over which questions were asked were not consistent; for example, injuries were asked about over the past 12 months, whereas some behaviours were asked about over the past 30 days or 7 days only. The duration of the behaviour is impossible to know from the responses. Descriptions of age groupings varied between countries and did not have specific cutoffs. We combined age ranges to ensure harmonisation across the surveys, but we were unable to present these more granularly. For example, in the Indonesia survey the lower and upper and lower cutoffs were '11 years old or younger' and '16 years and older'. Nevertheless, the surveys were done in secondary-school-age children; hence, the majority were between the ages of 13 and 17 years.

That the dataset did not have age as a continuous variable meant that it was not possible to tease out more subtle age differences in exposure that may occur on crossing an age milestone. For example, driving motorised vehicles or taking part in contact sports may increase on passing a country-determined age threshold. However, for females, going out unescorted may decrease with certain coming-of-age thresholds. Moreover, the dataset did not allow assessment of injury prevention strategies, for example helmet or seatbelt use, as too few countries included these questions at the time the assessments were done. The mechanisms and types of injuries and the associations studied were limited by the questions asked in the survey and were not exhaustive; for example, near-drowning was not included as a mechanism of injury. Furthermore, only the mechanism and type of the most serious injury was captured. This study relies on data from a retrospective self-reported survey, which may be affected by social desirability bias and recall bias. A prospective study that also captured school absences due to injuries may have overcome this limitation. This study looked at the associations between risk factors and outcomes but did not look at the effects of combinations of risk factors. We are aware that the literature shows a cumulative effect of some risk factors on childhood mental health [62,63], but assessing the cumulative effect on injuries goes beyond the remit of this study. The main age range covered was 13 to 16 years, whereas the WHO definition of adolescence is 10 to 19 years of age. Although some of the participants were below 13 or above 17 years old, no exact age was captured in the survey, and therefore we cannot be certain how representative our results are of adolescents of all ages. Finally, although several positive associations were seen with the category 'other' for both injury type and mechanism of injury, we were not able to identify what this category contained due to lack of detail in the survey data.

The strengths of this study include a comprehensive list of psychosocial factors and their association with serious injury occurrence, mechanism, and type. Additionally, the study involved a large sample size and a standardised procedure for participant selection. The analysis took into account survey design when calculating prevalence estimations and associations.

In this large multi-country study, we have shown that the association between injury and indications of psychosocial circumstances is complex. Serious injuries were strongly associated with multiple factors across categories of markers of risky behaviour, contextual factors, protective factors, indicators of poor mental health, and aggressive behaviour indicators. These findings reinforce the view that addressing injuries will require considering multiple SDGs, not just the specific injury SDG, 3.6. Multicomponent interventions to improve psychosocial circumstances will likely be needed to substantially reduce injury prevalence.

## Supporting information

**S1 STROBE Checklist. STROBE checklist.**
(PDF)

**S1 Fig. Forest plot showing the multivariable associations (as odds ratio [OR] and 95% confidence interval [95% CI]) between serious injury occurrence and markers of psychosocial circumstances, adjusting for country-level factors, demographic characteristics, and aggressive behaviour indicators.** Adjustment for non-response rate of participants was done as described in Methods. *p*-Values can be found in Table 3 (Model 2).
(TIF)

**S1 Text.** Table A: Number of responses of the total population and by occurrence of any serious injury in the last 12 months, for each of our variables for original and combined survey questions. Combined responses are shown highlighted yellow or orange. Table B: Characteristics of included countries and their populations (unweighted, unless stated otherwise). *Bangladesh did not have data on mechanism of injury, so was excluded from those analyses, but included in all others. Table C: Characteristics of the total population and by occurrence of any serious injury in the last 12 months, in total and by sex. Weights take into account the 2-stage study design and are also adjusted for the non-response rate of participants; see Methods for a full explanation. Table D: Multivariate analysis showing association between serious injury occurrence and individual characteristics, including aggressive behaviour indicators (Model 1), with income status (Model 2) and world region (Model 3) added separately. Table E: Characteristics of the total population and by occurrence of any serious injury in the last 12 months, in total:—Complete cases used in analysis of type of injury. Weights take into account the 2-stage study design and are also adjusted for the non-response rate of participants; see Methods for a full explanation. Table F: Characteristics of the total population and by occurrence of any serious injury in the last 12 months, by sex: Complete cases used in analysis of type of injury. Weights take into account the 2-stage study design and are also adjusted for the non-response rate of participants; see Methods for a full explanation. Table G: Characteristics of the total population and by occurrence of any serious injury in the last 12 months, in total: Complete cases used in analysis of mechanism of injury. Weights take into account the 2-stage study design and are also adjusted for the non-response rate of participants; see Methods for a full explanation. *Data on mechanism of injury were not captured in the Bangladesh survey. Table H: Characteristics of the total population and by occurrence of any serious injury in the last 12 months, by sex: Complete cases used in analysis of mechanism of injury. Weights take into account the 2-stage study design and are also adjusted for the non-response rate of participants; see Methods for a full explanation. *Data on mechanism of injury

were not captured in the Bangladesh survey.
(DOCX)

## Author Contributions

**Formal analysis:** Samiha Ismail, Justine Davies.

**Methodology:** Samiha Ismail, Maria Lisa Odland, Amman Malik, Misghina Weldegiorgis, Karen Newbigging, Margaret Peden, Mark Woodward, Justine Davies.

**Supervision:** Justine Davies.

**Writing – original draft:** Samiha Ismail, Maria Lisa Odland, Justine Davies.

**Writing – review & editing:** Samiha Ismail, Maria Lisa Odland, Amman Malik, Misghina Weldegiorgis, Karen Newbigging, Margaret Peden, Mark Woodward, Justine Davies.

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
