## [Editor Report · Decision Letter 0]

16 Feb 2021

Dear Dr Ismail, 

Thank you for submitting your manuscript entitled "The relationship between psychosocial circumstances and injuries in adolescents: an analysis of 87,269 individuals from 26 countries using the Global Schools-Based Student Health Survey" for consideration in PLOS Medicine’s Special Issue on Global Child Health.

Your manuscript has now been evaluated by the PLOS Medicine editorial staff as well as by the Special Issue Guest Editors, and I am writing to let you know that we would like to send your submission out for external peer review.

Kind regards,

Caitlin Moyer, Ph.D.

Associate Editor

PLOS Medicine

---

## [Decision Letter · Decision Letter 1]

6 May 2021

Dear Dr. Ismail,

Thank you very much for submitting your manuscript "The relationship between psychosocial circumstances and injuries in adolescents: an analysis of 87,269 individuals from 26 countries using the Global Schools-Based Student Health Survey" (PMEDICINE-D-21-00477R1) for consideration in PLOS Medicine’s Special Issue: Global Child Health: From Birth to Adolescence and Beyond.

Your paper was evaluated by a senior editor and discussed among all the editors here. It was also discussed with the special issue guest editors, and sent to three independent reviewers, including a statistical reviewer. The reviews are appended at the bottom of this email and any accompanying reviewer attachments can be seen via the link below:

[LINK]

In light of these reviews, I am afraid that we will not be able to accept the manuscript for publication in the journal in its current form, but we would like to consider a revised version that addresses the reviewers' and editors' comments. Obviously we cannot make any decision about publication until we have seen the revised manuscript and your response, and we plan to seek re-review by one or more of the reviewers. 

We expect to receive your revised manuscript by May 27 2021 11:59PM. Please email us (plosmedicine@plos.org) if you have any questions or concerns.

We look forward to receiving your revised manuscript. 

Sincerely,

Caitlin Moyer, Ph.D.

Associate Editor 

PLOS Medicine

plosmedicine.org

1. Abstract: Please structure your abstract using the PLOS Medicine headings (Background, Methods and Findings, Conclusions).

2. Abstract: Methods and Findings: Please provide summary demographics on the age/ranges of the adolescents. Please quantify the main results presented with both 95% CIs and p values. In the last sentence of the Abstract Methods and Findings section, please describe the main limitation(s) of the study's methodology.

3. Abstract: Conclusions: Please address the study implications without overreaching what can be concluded from the data; beginning with the phrase "In this study, we observed ..." may be useful. In the second sentence, please temper the wording slightly to avoid causal implications (“...programmes to target...are likely to reduce…”).

4. Main text: Please remove the sections “Funding” and “Declaration of interest” from the main text (this information should be included through the manuscript submission system).

5. Main text: Please include line numbers with the revised document.

6. Main text: Please ensure all abbreviations are fully defined at first use in the text (for example, DALYs in the Author summary).

7. Author summary: What did the researchers do and find?: Please clarify the third bullet point on protective factors- from the results presentation it seems as if greater numbers of friends, more understanding parents was associated with greater odds of injury, rather than an inverse relationship.

8. Author summary: What do these findings mean? Please revise the wording of the second bullet point to avoid implying causality (“Targeting...is likely to have the co-benefit of reducing…”).

9. Throughout: Please place citations within brackets prior to the sentence punctuation. For multiple references within brackets, please do not include spaces.

10. Methods: Did your study have a prospective protocol or analysis plan? Please state this (either way) early in the Methods section.

11. Methods: Please ensure that the study is reported according to the STROBE guideline, and include the completed STROBE checklist as Supporting Information. Please add the following statement, or similar, to the Methods: "This study is reported as per the Strengthening the Reporting of Observational Studies in Epidemiology (STROBE) guideline (S1 Checklist)."

12. Results: Page 11: Please avoid the use of language implying causality. For example, please rephrase sentences such as “Physical activity did not increase the odds of injury.” and “There was no protective effect of…” to avoid the use of causal language.

13. Results: Page 12: In the paragraph describing relationships with aggressive behavior variables, please replace “effect” with “association” in the following sentence (or similar): “...apart from attenuating the relationships between older age and odds of being injured, as well as attenuating the differences between geographical areas.”

14. Discussion: Please present and organize the Discussion as follows: a short, clear summary of the article's findings; what the study adds to existing research and where and why the results may differ from previous research; strengths and limitations of the study; implications and next steps for research, clinical practice, and/or public policy; one-paragraph conclusion.

15. Discussion: Page 20: Please provide some additional detail for this sentence: “Finally, although several positive associations were seen with the category “other”, lack of detail meant that it was not possible to know what this category contained.”

16. Figures and Tables: Please provide titles and legends for all figures (including those in Supporting Information files).

17. Table 1 and Table 3: Please include in the legend the explanation for the weighted percentages reported.

18. Table 2: Please clarify in the legend that Model 1 is the unadjusted model.

Comments from the reviewers:

Reviewer #1: I confine my remarks to statistical aspects of this paper. The general approach is fine, but I have some issues to resolve before I can recommend publication.

General: Was collinearity investigated? It seems likely with these variables.

p. 5 Please also say what % of adolescents are in LMICs. If it is approximately 90%, then the fact that 90% of deaths occur there is not really interesting.

 Give numbers of deaths for the various causes. To say something is a "leading cause" isn't very meaningful; even if deaths are very rare, something has to be the leading cause.

p. 10 How did you ``adjust for study design"? That you used certain variables is not sufficient information. Did you use robust standard errors? A multilevel model? GEE? Something else?

 It is good that you used forced entry.

 Rather than list a version of RStudio, it would be good to list a version of R. RStudio is just an editor. (After all, you don't list which version of Word you used).

Figures: The font is too small to be readable on a printout. One way to help fix this is to merge the two left-most columns and simply indent the variable names by a few spaces. I'm not sure if that will be enough, but it's a start.

Peter Flom

Reviewer #2: This study examined psychosocial factors associated with serious injury occurrence, mechanism, and type in adolescents. Its sampling methodology, measurement of the variables, and writing are appropriate. 

My concerns about the article mostly are:

1. Although there are many correlations in this study the theoretical support of the manuscript is lacking, what are the motivations of the study based in the behavioral and psychology literature? Lacking theoretical perspectives, the model itself cannot answer the why manifests these associations.

2. Describe about psychosocial circumstances is not clear, please clarify.

3. Family and school factors are important contextual variables, please answer why did not included them in analysis.

4. This multi-stage random sampling design, but statistic analysis did not account for multiple stages of sampling stratification and clustering, which may lead to some errors. I suggest conduct re-analyze using complex sample analysis methods.

5. The discussion is merely a reiteration of results and comparison with some published studies. In-depth explanation of the findings is necessary.

6. The manuscript appeared messy in many associations describes, proper organization and concentration are necessary.

Reviewer #3: This is a well written paper on important, if not complex, analyses to further understanding on how best to prevent injuries among youth. It is strongly presented as is. Just some thoughts are offered and only minor recommended revisions.

While appreciating the numerous variables to explain and the need for appendices and referral to other publications, one aspect important to explain within this manuscript is the age groupings. This does not appear to be addressed (besides the need for upper and lower cut-offs and lack of age range). Was this a grouping agreed by authors and/or how was this determined?

In different country socio-cultural contexts, there could perceivably be large differences between 14 years and 15 years for example in terms of exposures, particularly by gender; for example, acceptability of young males versus females being out unchaperoned at night, taking part in contact sports, official working age or legal age for riding motorised vehicles (such as mopeds); appreciating there are controls by country. While the limitations section was particularly strong and appreciated, the lack of ability to measure or control for such other types of 'exposures' to injury risk might also be acknowledged.

The addition of the aggressive behaviour indicator variables did not (essentially) change the main model and all held significant associations. Did you explore the proportions of variance explained by the models and might these additions have increased the explained variance?

Thinking through the curiosity of lack of anticipated protective factor associations, I wonder if this might relate to the dominant low-to-middle income country participation, with most of the associated research from high-income countries, as acknowledged by the authors. It is possible there is more to understand about the full wording of the questions and language translation (including whether translations were both from English and back to English?). Just "having more" friends and "supportive parents" out of context may be blurred as protective within (sub)cultures with strong criminogenic tendencies. That is, more friends and parents supportive of criminogenic behaviours would then be another form of increased exposure to injury and would cancel out expected protective associations. Within offender management literature, reducing contact with criminogenic family and friends is one of the proven success factors to reduce reoffending (see for example, evidence-based work led by James Bonta and Donald Andrews on the Risk-Need-Responsivity model). 

Minor

Abstract: Citing unintentional injury statistics here, and then again in the Introduction, made it not entirely certain whether these were the main focus rather than both intentional and unintentional until reading the Methods. This could be more explicit.

Abstract: why is "<37%" included rather than the actual rounded %?

Last point on p4, "All available evidence points to psychosocial factors such as bullying, drug taking, or poor mental health are strongly associated…", should be "as strong associated"?

Introduction first sentence is missing a reference.

Methods: Noting whether self-completion of the survey was at school in all contexts or perhaps independently on-line for example would add further contextual understanding.

Results: There was inconsistent reporting of the p value findings in terms of conventions of <0.05, <0.01, <0.001, to reporting "=" to rounded values. I noted only one case where p=0.05 was appropriate (last line page 12), otherwise it would be less distracting to consistently apply the 'less than' convention.

Page 11 paragraph 4, "Considering markers of risky behaviour, ever having used drugs was significantly associated with injury (OR 2.08, 95% CI 1.73 - 2.49, p< 0.001), but the relationships between smoking or alcohol and injury were not consistent": use of "were not consistent" is ambiguous as to whether this refers to the same as 'ever used drugs' or consistent between alcohol and smoking, rather than a lack of dose-response type associations for these variables - which I assume this was meant to convey. Perhaps just "were inconsistent" would be better here. 

There is also some inconsistency in explaining that the cited odds examples in brackets are for the highest dose/level only for some of the variables with multiple levels for which all levels were significant (compared to the reference value). For example, this is stated as an example "for most of the time or always" on last line page 11, but not elsewhere. While it might be somewhat repetitive to state this on every occasion, an overarching statement could be made that the highest dose/level statistics are included as examples in brackets, when applicable (adding "e.g.," to the bracketed examples).

Discussion: there were a few typos to spell check in this section (e.g., top page 16 "bulling").

Top of page 18: I was anticipating a comment about 'but not for considering suicide only'?

Then to end the paper, this somehow seemed to lack a strong conclusion section, despite the many strength and insights on offer. The end paragraph was more focused on implications but could have offered some more summary statements also. Note that 'SDGs' is not first explained or referenced before use here.

Note also that the copies of the figures within the PDF were almost illegible (very blurry). In reviewing, rather than persevere to try to check all of these, the main focus was on the tables (besides noting the patterns that are helpful to have visually given the many numbers in lengthy tables).

[LINK]

---

## [Decision Letter · Decision Letter 2]

28 Jun 2021

Dear Dr. Ismail,

Thank you very much for re-submitting your manuscript "The relationship between psychosocial circumstances and injuries in adolescents: an analysis of 87,269 individuals from 26 countries using the Global Schools-Based Student Health Survey" (PMEDICINE-D-21-00477R2) for consideration in PLOS Medicine’s Special Issue: Global Child Health: From Birth to Adolescence and Beyond.

I have discussed the paper with my colleagues and the academic editor and it was also seen again by three reviewers. I am pleased to say that provided the remaining editorial and production issues are dealt with we are planning to accept the paper for publication in the journal.

[LINK]

We look forward to receiving the revised manuscript by Jul 05 2021 11:59PM.   

Sincerely,

Caitlin Moyer, Ph.D.

Associate Editor

PLOS Medicine

plosmedicine.org

Requests from Editors:

1. Abstract: Background: If possible, please be more specific when describing “Large numbers of adolescents…”

2. Abstract: Line 46: Please clarify what is meant by “geography” in terms of the country-level factors for which you adjusted.

3. Abstract: Conclusions: Line 68: We would suggest “...but we note the relationships are likely to be complex and our findings do not inform causality.” or similar, to clarify.

4. Abstract: Line 69: We suggest changing “indicate” to “suggest” and at line 70-71, we suggest changing to read “...in particular those concentrating on reducing bullying and drug use and improving mental health.”

5. Introduction: Line 155: We suggest “..about each adolescent’s health…” or similar.

6. Methods: Line 186: Please change to “...we agreed on variables…” or similar.

7. Methods: Line 191-193: Thank you for the clarification regarding the planning of your analyses. Please mention in the text the variables that were intended to be included, but could not be included due to limited availability: “...hence some variables that were listed in the core set, but found to have limited availability, were discarded before agreeing on the final set for inclusion.”

8. Methods: Line 218-219: Please change to “Their inclusion was agreed on through discussion amongst authors.” or similar.

9. Methods: Early in the methods, please add the following statement, or similar, to the Methods: "This study is reported as per the Strengthening the Reporting of Observational Studies in Epidemiology (STROBE) guideline (S1 Checklist)."

10. Methods: Please provide the name(s) of the institutional review board(s) that provided ethical approval for the study.

11. Results: Line 297-298: Please consistently provide the results, with 95% CIs and p values for all major results described in the text, even if not statistically significant (“Physical activity was not associated with an increased odds of injury.”

12. Results: Throughout this section (as well as any tables/figures) please report exact p values, unless p<0.001.

13. Results: Line 334: Please change this to “...ever having used drugs…” and please also make this change at line 365.

14. Results: Line 366: Please clarify if this is meant to indicate “...being injured by something falling…” or similar.

15. Discussion: Line 402: We suggest changing to “...these relationships may be consistent across world regions.” or similar.

16. Discussion: Line 413: We suggest changing to “... suggests that improving the psychosocial environment for adolescents across multiple country settings could have benefits for prevention of injury.”

17. Discussion: Line 417: Please change “...were not protective of…” to “...were not associated with…”

18. Discussion: Line 449-451: Please revise to: “... being struck by an object…”

19. Figure 1 and Figure 2: In the left column, some of the category names are cut off (e.g. Number of days smoked in...); Number of days of alcohol in…). Please add p values to the figure, in addition to the 95% CIs or please indicate these are presented in the tables. Please note in the legend the variables adjusted for in the analysis, or if these are the results of the unadjusted analysis. If adjusted, please also ensure that the results of the unadjusted analysis are presented in the manuscript (at least in a table).

20. Figure 2: Please consider splitting the table up, or reformatting, as the font size seems difficult to read when sized to fit on a single page.

21. Table 2, Table 4, Table 5: In the legends, please note the variables adjusted for in each model, and for the analyses in Tables 4 and 5, please also include the unadjusted results. Please present the actual p value, except where p<0.001. Instead of NS, please indicate the p values.

22. References: Please check each reference for “Vancouver” style (see our guidelines at: https://journals.plos.org/plosmedicine/s/submission-guidelines#loc-references), and please check all journal title abbreviations. For example, in reference 48, The British Journal of Educational Psychology should be Br J Educ Psychol.

Comments from Reviewers:

Reviewer #1: The authors have addressed my concerns and I now recommend publication.

Reviewer #2: The authors adequately addressed my concerns. I think this manuscript may be accepted.

Reviewer #3: The authors have thoroughly addressed the reviewer comments.

I suggest the statement in their first reply to Reviewer #1, that "none of the variables met the criteria for collinearity" is added to the manuscript. This in part is important to the findings regarding explained variance. I personally think the statistics and comments made in this regard are useful to the reader, and could be stated within the text (rather than requiring an additional table), but am happy to defer to the Editor.

Some minor grammatical revisions might also be addressed in copyediting 

48-50 "from lower-middle income country (81.93%), or from East Asia and the Pacific (66.83%)" - use of "or" is incorrect as the percentages total more than 100.

88 "We have quantified..." - delete "have" for consistent grammatical tense with the other points.

109 "Focusing on psychosocial circumstances of adolescence might have the co-benefit of reducing injury prevalence in adolescence,..." is non-specific, rather suggest adding context, as referred to elsewhere of "multifactorial programmes".

Table legend "Weights take into account the 2-stage study design and is also adjusted for the non-response rate of participants, see the methods section for a full explanation." Change "is also adjusted" to "also adjust"?

[LINK]

---

## [Editor Report · Decision Letter 3]

8 Jul 2021

Dear Dr Ismail, 

On behalf of my colleagues and the Academic Editor, Kathryn Yount, I am pleased to inform you that we have agreed to publish your manuscript "The relationship between psychosocial circumstances and injuries in adolescents: an analysis of 87,269 individuals from 26 countries using the Global Schools-Based Student Health Survey" (PMEDICINE-D-21-00477R3) in PLOS Medicine’s Special Issue: Global Child Health: From Birth to Adolescence and Beyond.

In addition, please complete the following editorial requests: 

1. Title: Please capitalize the first letter of the subtitle: "The relationship between psychosocial circumstances and injuries in adolescents: An analysis of 87,269 individuals from 26 countries using the Global Schools-Based Student Health Survey"

2. Data availability statement: Please check that the following link is correct: www.who.int/ncds/surveillance/gshs/ both here as well as at line 282-288 in the Methods.

3. Abstract: At line 60, please replace p<0.01 with p=0.007, and at line 61, please replace p<0.05 with p=0.036 (if these are the correct values).

4. Author summary: Line 96-97: Please revise to "...having close friends or understanding family members or guardians were not significantly associated with the occurrence of serious injury."

5. References: For Reference 27, please add the semicolon after the publication year. For Reference 33, please add the period after the journal title.

PRESS

Sincerely, 

Caitlin Moyer, Ph.D. 

Associate Editor 

PLOS Medicine